

# Exploring the influence of self-identification on perceptual judgments of physical and social causality

Michele Vicovaro[1], Francesca Squadrelli Saraceno[2] and Mario Dalmaso[2]

[1] DPG, University of Padova, Padova, Italy
[2] DPSS, University of Padova, Padova, Italy

Corresponding authors
Michele Vicovaro,
michele.vicovaro@unipd.it
Mario Dalmaso, mario.
dalmaso@unipd.it

## ABSTRACT

People tend to overestimate the causal contribution of the self to the observed outcome in various situations, a cognitive bias known as the 'illusion of control.' This study delves into whether this cognitive bias impacts causality judgments in animations depicting physical and social causal interactions. In two experiments, participants were instructed to associate themselves and a hypothetical stranger identity with two geometrical shapes (a circle and a square). Subsequently, they viewed animations portraying these shapes assuming the roles of agent and patient in causal interactions. Within one block, the shape related to the self served as the agent, while the shape associated with the stranger played the role of the patient. Conversely, in the other block, the identity-role association was reversed. We posited that the perception of the self as a causal agent might influence explicit judgments of physical and social causality. Experiment 1 demonstrated that physical causality ratings were solely shaped by kinematic cues. In Experiment 2, emphasising social causality, the dominance of kinematic parameters was confirmed. Therefore, contrary to the hypothesis anticipating diminished causality ratings with specific identity-role associations, results indicated negligible impact of our manipulation. The study contributes to understanding the interplay between kinematic and non-kinematic cues in human causal reasoning. It suggests that explicit judgments of causality in simple animations primarily rely on low-level kinematic cues, with the cognitive bias of overestimating the self's contribution playing a negligible role.

## INTRODUCTION

Philosophers have characterised causality as the 'cement of the universe' (*Mackie, 1980*). Our perception of the world largely manifests itself as a meaningful sequence of interconnected events, attributing this coherence to our ability to readily infer cause-effect relationships between them. This inclination to perceive causal connections is pervasive, leading individuals to infer cause-effect relationships even when events are unrelated—an occurrence termed the 'illusion of causality' (*e.g.*, *Blanco, 2017*).

Given the pivotal role of causality in everyday life, psychologists have explored it extensively from various perspectives. Notably, a significant finding in decades of

psychological research on causality is that its representation does not solely rely on cognitive representations of statistical concepts like correlation or the Bayes theorem. Instead, causality can be visually perceived from simple stimuli, as demonstrated by *Michotte (1946/1963)* in the *launching effect*, where the motion of object *B* appears caused by the impact of object *A* (Fig. 1A), even though observers are aware that *A* and *B* are nothing else than geometric shapes moving on a uniform background. This visual phenomenon falls under the category of physical (or mechanical) causality, as it conveys the impression that *B* moves due to energy transmitted to it through physical contact with *A* (*Hubbard, 2013a*, *2022*).

Researchers in causal perception have questioned whether it is genuinely visual or a product of high-level cognitive processes relying on past experiences (*Rips, 2011*). Empirical evidence supports the former, as the effect depends on subtle manipulations of visual scene features like relative trajectories, speeds, distances, and timing (*Kiritani, 1999*; *Scholl & Tremoulet, 2000*; *Choi & Scholl, 2004*; *Scholl & Nakayama, 2004*; *Choi & Scholl, 2006a*, *Bae & Flombaum, 2011*). The strict dependence on visual scene features allows the perception of the launching effect even when a cause-effect relationship is counterintuitive. Moreover, the perception of the launching effect can influence low-level visual properties, demonstrating an interaction between the effect and basic visual processes (*Scholl & Nakayama, 2004*; *Parovel & Casco, 2006*; *Buehner & Humphreys, 2010*; *Kim, Feldman & Singh, 2013*; *Vicovaro, Battaglini & Parovel, 2020*; *Bechlivanidis et al., 2022*). It also exhibits distinct behavioural effects (*Badler, Lefèvre & Missal, 2010*, *2012*; *Rolfs, Dambacher & Cavanagh, 2013*; *Kominsky et al., 2017*; *Moors, Wagemans & de-Wit, 2017*; *Kominsky & Scholl, 2020*) and neurophysiological effects (*Fonlupt, 2003*; *Fugelsang et al., 2005*; *Roser et al., 2005*) that can be measured without explicitly asking participants about their causal impressions. Additionally, sensitivity to mechanical causality emerges early in individual development (*Leslie & Keeble, 1987*; *Oakes & Cohen, 1990*; *Rakison & Krogh, 2012*).

## From 'physical' to 'social' causality

In addition to the launching effect, diverse manifestations of mechanical causality encompass enforced disintegration, bursting (*White & Milne, 1999*), shattering (*Hubbard & Ruppel, 2013*), and bouncing (*Vicovaro, Brunello & Parovel, 2023*). Causal impressions may also arise from alterations in the size, shape, or colour of *B* subsequent to its interaction with *A* (*Bechlivanidis, Schlottmann & Lagnado, 2019*). These instances only represent a fraction of the broader scope of causal impressions (for an extensive review on various visual impressions of causality, refer to *Hubbard, 2013a*, *2013b*). *Kanizsa & Vicario (1968)* introduced a transformative adjustment in the visual display that gives rise to the launching effect, where object *B* initiates motion prior to contact with *A*, as depicted in Fig. 1B. Under specific conditions, this temporal shift induces the impression of an intentional reaction, imparting the observer with the percept that *A* is attempting to apprehend *B*, who, in turn, appears to evade capture. This intentional reaction effect stands in stark contrast to the launching effect, portraying *A* and *B* as entities capable of intentional actions and reactions. The associated causality impression is termed social or psychological causality, reflecting the apparent influence of social and psychological
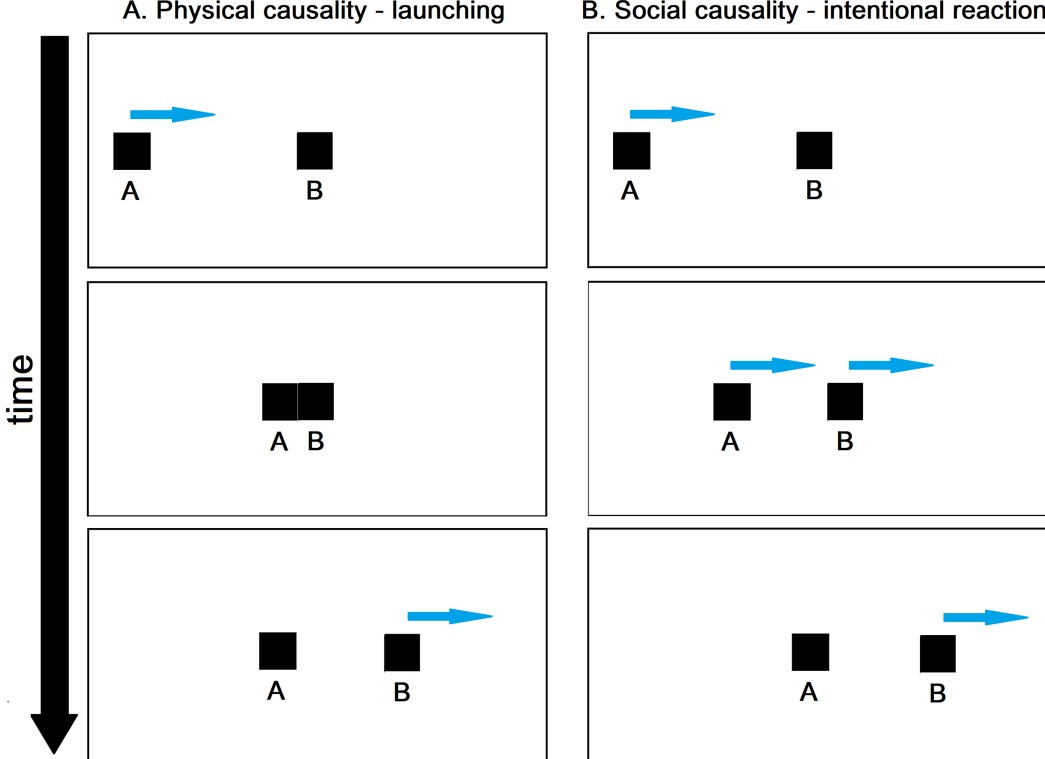

**Figure 1 Representation of animations associated with the perception of physical and social causality.** (A) Three frames depicting the launching effect (physical causality). (B) Three frames depicting an 'intentional reaction' effect (social causality). Arrows indicate the direction of the object motion and are not part of the animation sequence, as well as letters.

processes on object motions. *Kanizsa & Vicario*'s *(1968)* initial findings, predominantly derived from expert observers, have been corroborated in more extensive studies involving naïve adults (*Schlottmann et al., 2006*). Sensitivity to intentional reactions extends beyond adults and has been observed in children and infants as well (*Schlottmann & Surian, 1999*; *Schlottmann et al., 2002*; *Schlottmann, Surian & Ray, 2009*).

Significant distinctions emerge between stimuli eliciting the launching and intentional reaction effects. In the temporal dimension, we define delay as the interval from *A–B* contact to *B*'s motion onset. Optimal conditions for a launching impression necessitate a null or slightly positive delay (not exceeding 70 ms), whereas intentional reaction impressions necessitate a negative delay (*i.e.*, *B* starts moving prior to the contact with *A*). *Kanizsa & Vicario (1968)* demonstrated that a delay between −400 and −50 ms with *A* moving at 4 cm/s, induces intentional reaction impressions. They contended that for delays smaller than −400 ms, the perceived distance between *A* and *B* becomes too extensive to facilitate the perception of a causal link between their motions. Notably, the intentional reaction appears independent of *A*'s subsequent pursuit or cessation after *B*'s departure. Besides the temporal aspects, the relation between *A*'s and *B*'s speeds is crucial for distinguishing between the launching impression and intentional reaction impression. Launching impressions require *B*'s speed to be equal or slightly smaller than *A*'s, whereas for an intentional reaction impression, *B* should move equal to or faster than *A*.

Attempts to broaden the investigation of social causality into bi-dimensional motion, encompassing shapes moving across a plane (*Bassili, 1976*; *Blythe, Miller & Todd, 1999*; *Falmier & Young, 2008*), reveal overarching principles: (1) one of the two objects (the chaser) should be perceived as initially approaching another object (the chasee); (2) when the chaser is perceptually close to the chasee, the latter should either increase its speed and/ or change direction, momentarily enlarging the distance with the chaser; (3) shortly after the speed and/or motion direction change of the chasee, the chaser's motion adjusts accordingly, causing the distance between the two objects to resume decreasing. This sequential pattern can be iterated multiple times.

More recent investigations (*Gao, Newman & Scholl, 2009*; *Gao, McCarthy & Scholl, 2010*; *Gao & Scholl, 2011*) have delved into the visual dynamics of chasing, employing elaborate scenarios involving more than two objects—comprising a chaser, a chasee, and a variable number of distractors—moving on a plane for extended durations (10 s or more), with frequent changes in motion direction. *Gao, Newman & Scholl (2009)* identified that optimal clarity in chasing impressions occurred when the chaser's and chasee's motion directions were identical. That is, when after a direction change by the chasee, the chaser promptly aligned its motion direction. Chasing impressions were also observed when the maximal angular deviation of the chaser's direction relative to the chasee's direction was within +/− 30°, while wider deviation angles led to a significant decrease in chase detection accuracy. These findings underline the pivotal role of consistent motion direction alignment between the chaser and chasee in shaping perceptions of social causality in chasing scenarios. The study by *Gao, McCarthy & Scholl (2010)* proposes a complementary perspective, suggesting that the perceived orientation of objects within the scene serves as a potent social cue. This alone can suffice to generate a compelling chasing impression. Intriguingly, the study introduced the concept of a 'wolfpack effect', wherein a group of moving distractors, referred to as 'wolves', maintained continuous orientation towards a target stimulus, or the 'sheep'. This configuration created a persuasive impression of 'wolves' chasing the 'sheep', even when the relative motions of the 'wolves' and the 'sheep' did not align with a typical chase-escape interaction.

Concurrent with investigations into the launching effect, recent research on the visual perception of social causality has centred on unravelling the role of low-level perceptual processes *vs.* high-level cognitive inferences. Numerous studies suggest that, akin to the launching effect, the visual perception of chasing is contingent upon subtle manipulations of low-level features within the scenario, exhibiting measurable implicit behavioural effects (*Gao, Newman & Scholl, 2009*; *Gao, McCarthy & Scholl, 2010*; *Gao & Scholl, 2011*; *Scholl & Gao, 2013*; *van Buren, Uddenberg & Scholl, 2016*; *van Buren, Gao & Scholl, 2017*; *Parovel & Guidi, 2020*) and neurophysiological effects (*Schultz et al., 2005*). These effects persist even when tasks do not necessitate explicit judgments regarding the presence of social causality, chasing, or animacy. This substantiates the proposition that impressions of social causality result from automatic low-level processes such as shape, depth, and mechanical causality perception.

## Explicit judgments of physical and social causality: perceptual and post-perceptual processes

Despite the well-established presence of visual impressions related to physical and social causality, discerning the specific contributions of low-level perception and post-perceptual reasoning to participants' responses proves challenging. When participants are presented with stimuli like those in Figs. 1A and 1B and are prompted to provide causal judgments through verbal reports or numerical ratings, disentangling the influence of low-level perception from post-perceptual reasoning is virtually impossible (*Schlottmann, 2001*; *Choi & Scholl, 2006b*; *Schlottmann et al., 2006*; *Gao, Newman & Scholl, 2009*; *Scholl & Gao, 2013*). In essence, explicit causality reports are typically influenced by variable combinations of low-level visual processes relying on kinematic cues and high-level cognitive processes. If causal judgments were solely driven by genuine visual impressions based on kinematic cues, explaining their substantial inter-individual variability (*Gemelli & Cappellini, 1958*; *Schlottmann & Anderson, 1993*; *Falmier & Young, 2008*) and their dependency on task characteristics and demands would be challenging. For example, studies since *Michotte*'s *(1946/1963)* work have shown that the launching effect can be disrupted by introducing a significant delay between *A–B* contact and *B*'s motion onset. Despite this, animations with long delays may still receive high causality ratings based on the stimuli participants encounter during the experimental session (*Powesland, 1959*; *Brown & Miles, 1969*; *Schlottmann et al., 2006*; *Badler, Lefèvre & Missal, 2012*; *Bechlivanidis, Schlottmann & Lagnado, 2019*; *Deodato & Melcher, 2022*). This context-dependency is attributed to the fact that the context shapes the subjective meaning of causality, influencing participants' explicit ratings. Furthermore, a study by *Vicovaro (2018)* showed that, in a scenario featuring simulated collisions between 3D objects seemingly composed of distinct materials, explicit causality ratings were primarily shaped by observers' intuitive understanding of collision physics rather than by visual impressions of launching.

In the realm of social causality judgments, there is substantial evidence that these judgments are significantly influenced by post-perceptual cognitive processes. For instance, when simple chasing sequences are embedded in a complex animation depicting various social interactions between abstract geometric shapes, the interpretation of the simple sequences relies on the broader meaning conveyed by the animation as a whole (*Heider & Simmel, 1944*). Judgments of social causality are also influenced by observers' cultural background (*i.e.*, individualistic or collectivistic culture; *Morris & Peng, 1994*), beliefs about the animate or inanimate nature of objects in the scene (*Schlottmann et al., 2006*; *Falmier & Young, 2008*), and semantic information provided before the target animation, such as the personality (*Shor, 1957*) or emotional status (*Thayer & Schiff, 1969*) of actors. Notably, even in infancy, perceptions of mechanical and social causality are shaped by perceived characteristics of the involved actors, such as their animation status (*Saxe, Tenenbaum & Carey, 2005*; *Saxe, Tzelnic & Carey, 2007*; *Muentener & Carey, 2010*) and the potential for communication between the actors (*Kosugi & Fujita, 2002*).

Delving into the purely perceptual mechanisms underlying our comprehension of physical and social causality is a commendable scientific pursuit. However, it is essential to recognise that artificially isolating visual processes from post-perceptual cognitive processes does not enable a comprehensive understanding of explicit causality judgments. As mentioned earlier, these judgments are influenced by both visual and higher-level cognitive processes, including contextual effects related to the type of stimuli that are presented within an experimental session. The amalgamation of visual impressions of causality, stemming from kinematic cues, with information derived from an individual's accumulated knowledge of cause-effect relationships, shaped by past experiences, is not only an empirically established fact in explicit causality judgments, but is also adaptive. Compared to inferences based solely on automatic and cognitively impenetrable visual impressions driven by kinematics, this integration enables individuals to offer more comprehensive, realistic, and predictive causal assessments (*Shultz, 1982*; *Shultz et al., 1986*; *Schlottmann, 1999*, *2001*). Despite the apparent importance of this consideration, the specific nature of non-perceptual, higher-level information that can integrate with kinematic information to formulate explicit judgments of causality remains largely unknown.

## Outline of the present study

In a broad sense, this study aims to investigate the interplay between kinematic information concerning physical and social causality and non-visual information related to the identity of the scene's actors. We employed a novel experimental task, wherein two shapes engaged in animations depicting physical or social causality were initially linked either to the self or to a stranger. Our primary objective was to investigate how information about the identity of objects involved in a physical or social causality scenario integrates with kinematic information, thereby influencing participants' explicit reports of causality.

Before delineating our specific hypotheses, it is pertinent to delve into two distinct yet interconnected topics: the differentiation between the roles of agent and patient in causal interactions, and the concept of self as a causal agent. Initiating our exploration, we recognise that the perception of causality manifests a pronounced asymmetry, wherein one of the interacting objects is perceived as an agent, and the other as a patient. This assignment of agent and patient roles relies on kinematic cues. As discussed by *White (2006a)*, in simple animations depicting physical causality, the object initiating movement or moving more rapidly is construed as the agent, while the initially stationary or slower-moving object is construed as the patient (*e.g.*, in the animations depicted in Fig. 1A, object *A* is perceived as the agent, and object *B* as the patient). Additionally, the object displaying the most visible outcome post-interaction often takes on the role of the patient. This agent-patient distinction permeates causal perception and reasoning (*White, 2006a*) and is associated with the psychological phenomenon known as causal asymmetry (*White, 2006b*, *2007*). In the realm of causal perception, causal asymmetry denotes the phenomenon where causality is predominantly perceived as unidirectional. The motion of the patient is almost exclusively attributed to the agent, and the patient's contribution to its own motion is largely overlooked. This contrasts starkly with Newtonian mechanics,

characterised by symmetrical forces, where objects involved in a dynamic interaction, equally contribute to the resulting motions. For example, if the objects in Fig. 1A were real physical entities, the mass of *B* would be no less influential than the mass of *A* in determining the post-collision motions of both *A* and *B*. However, at a perceptual level, *B*'s motion is phenomenally linked to the contact with *A*, while *B*'s contribution to its own motion appears negligible[1]. It is noteworthy that, although *White*'s *(2006a*, *2006b*, *2007)* discussions on causal asymmetry did not cover stimuli configurations eliciting social causality perception, the differentiation between a 'chaser' and a 'chasee' clearly illustrates an asymmetry in causal roles within those configurations. Moving to the second core concept requiring discussion, we consider the association between the concept of self and the concept of causal agency. Multiple lines of inquiry suggest that individuals tend to perceive themselves as causal agents. According to *White (2006b*, *2007)*, the causal asymmetry is linked to the fact that, in most daily perceptual-motor interactions with physical objects, individuals perceive themselves as active agents planning and anticipating the outcome of an intended action on a physical object, moving their muscles to generate the desired outcome. Because the objects on which these volitional acts are applied generally comply with the individual's intended actions, the contribution of these objects to the observed outcome often goes undetected. In the domain of causal reasoning, the illusion of control denotes the tendency for individuals to overestimate the influence of their skills and behaviour on various events (*Langer, 1975*; *Alloy & Abramson, 1979*; *Thompson, Armstrong & Thomas, 1998*; *Thompson, 1999*). Essentially, people may inaccurately perceive their actions as the causal determinant of an observed outcome, even in situations where the outcome is purely the result of chance, with no actual causal connection between their behaviour and the observed result. This phenomenon underscores the intimate relationship between the concepts of self and causality, giving rise to cognitive biases (as well as superstitions).

This study aimed to investigate whether, and how, associating the self or a stranger's identity with shapes playing the roles of agent or patient in animations depicting physical or social causality could influence causal judgments. To elucidate the logic of the two experiments, a brief summary of the experimental procedure is provided, omitting several methodological details. The two experiments began with a learning phase, where participants established arbitrary associations between their identity (self) and a geometric shape (*e.g.*, circle), as well as between a stranger's identity and another shape (*e.g.*, square). Subsequently, participants viewed animations showing instances of physical or social causality, with the self-related shape as the agent and the stranger-related shape as the patient. After each animation, participants were asked: 'How much did you cause the motion of the stranger?' In a subsequent block of animations, the roles were reversed, with the self-related shape as the patient and the stranger-related shape as the agent. After each animation, participants were asked: 'How much did the stranger cause your motion?' If the concept of the self as an active agent converges with visual impressions of causality, we anticipated lower causality ratings when the stranger is cast in the role of the agent and the self assumes the role of the patient, compared to the reverse scenario. Indeed, if the self is recognised as an active agent beyond basic visual cues, some inherent source of movement

---

[1] *White (2006b*, *2007)* discusses two types of causal asymmetries. One is presented in the main text, while the other is detailed here. Specifically, individuals tend to neglect or to underestimate the causal effect of the patient on the agent. If the objects in Fig. 1A were real physical entities, *A*'s post-collision motion (or lack thereof) would be equally caused by the collision with *B* as *B*'s post-collision motion. However, observers do not perceive *B* as the cause of *A*'s post-collision behaviour. This causal asymmetry is also evident in a perceived force asymmetry, wherein the force exerted by the patient on the agent is largely underestimated compared to the force perceived from the agent to the patient (*White, 2012*; *Hubbard & Ruppel, 2013*, *2017*).

should be attributed to the self, even when it takes on the role of the patient in a causal interaction. Simultaneously, the causal role of the agent should be emphasised when associated with the self as compared to when associated with the stranger. This observation would signify the integration of low-level kinematic cues with higher-level self-identification mechanisms in shaping explicit judgments of physical and social causality.

Concluding this introduction, it is crucial to highlight a methodological consideration. To the best of our knowledge, the concept of associating shapes involved in a causal interaction with different identities has not been explored in the literature on causal perception; rather, we draw inspiration from the literature on self-prioritisation. The well-established psychological phenomenon of self-prioritisation underscores the prioritisation of self-relevant information over other-relevant information (*Sui & Humphreys, 2015*; *Cunningham & Turk, 2017*). While the present study does not directly delve into the self-prioritisation effect, we refer to this literature to support our assumption that self and stranger identities can be arbitrarily linked to neutral geometric shapes. In a seminal study in this research domain (*Sui, He & Humphreys, 2012*), participants associated themselves, a friend, and a stranger with three arbitrary geometric shapes (a circle, a triangle, and a square). In subsequent matching tasks, participants discerned whether shape-label pairs were correct or incorrect based on the learnt associations. The results revealed faster and more accurate responses when the label 'you' was paired with the self-related shape (*e.g.*, circle + you) compared to trials with other shape-label associations (*e.g.*, square + you, square + stranger, triangle + friend). Subsequent studies have supported the idea that self and stranger can be associated with various stimuli (*e.g.*, *Frings & Wentura, 2014*; *Schäfer et al., 2016*; *Stein, Siebold & van Zoest, 2016*; *Woźniak & Knoblich, 2019*; *Vicovaro, Dalmaso & Bertamini, 2022*). Within the scope of the present study, the crucial observation is that a simple learning phase is adequate to establish an arbitrary association between two identities (self and stranger) and two geometric shapes (circle and square).

## Experiment 1

This first experiment focused on animations featuring horizontally moving objects. Informed by findings from earlier studies (*Michotte, 1946/1963*; *Kanizsa & Vicario, 1968*), kinematic parameters of the animations were adjusted to evoke perceptions of either physical causality or social causality. Additionally, a subset of animations was intentionally devoid of any discernible causal impression. The objective was to investigate whether the linkage between identity (self *vs.* stranger) and causal role (agent *vs.* patient) could influence causal judgments specifically in the context of physical and/or social causality impressions.

## MATERIALS AND METHODS

### Sample size determination

Our primary hypothesis posits that causality ratings will be higher when the agent is associated with the self and the patient with the stranger, compared to the reverse association. Unfortunately, we cannot draw upon the results of previous studies to estimate

the expected effect size, as, to the best of our knowledge, this study represents the first attempt to test this hypothesis. We opted for a hypothetical small-to-medium effect size of $d = 0.4$, aiming for a balance between parsimony and scientific relevance. This effect size pertains to the overall comparison between causality ratings for the self-agent/stranger-patient association and those for the stranger-agent/self-patient association, irrespective of other manipulated variables.

Sample size determination and subsequent analyses were conducted in R (*R-Core-Team, 2023*). Specifically, the sample size was estimated using the *pwr.t.test* function within the *pwr* package (*Champely, 2020*) for a unidirectional paired-sample *t*-test with $d = 0.4$ and *power* = 0.9, resulting in an estimated sample size of $N = 54.9$. For practical reasons, we decided to test a slightly larger sample, acknowledging the potential exclusion of some participants post-experiment if they failed to meet specific inclusion criteria. We stopped at $N = 64$, at the end of a booking session.

## Participants
Sixty-four naïve students (*Mean age* = 24 years, *SD* = 4.08 years, 38 females) took part in the experiment in exchange for course credits. All participants had normal or corrected-to-normal vision and provided written informed consent before the beginning of the experiment. The study was approved by the Ethics Committee for Psychological Research at the University of Padova (approval number: 3455) and was in accordance with the ethical standards of the 1964 Helsinki Declaration.

## Stimuli, apparatus, and procedure
Data were collected on a PC running PsychoPy (*Peirce et al., 2019*). Participants sat about 57 cm from the monitor (1,024 × 768, 60 Hz). The background colour was set to medium grey (RGB = (0.502, 0.502, 0.502)), whereas text and shapes were set to black. The entire experiment consisted of six phases (see Fig. 2A). For experiment code see the Data Availability section. A summary of the six phases is provided here, whereas a detailed description of each phase will be provided in subsequent paragraphs. First phase: participants received verbal instructions to establish an association between two shapes (circle and square) and two identities (self and stranger). Second phase: a shape-identity matching task was administered to reinforce the acquired shape-identity association and assess its learning and retention. Third phase: participants were randomly presented with 72 animations, preceded by six practice animations. The objective was to evoke physical or social causality impressions of varying strength. Half of the participants observed the self-associated shape as the agent and the stranger-associated shape as the patient, while the other half experienced the opposite identity-role association. After each animation, participants rated causality. Fourth phase: a repetition of the matching task from the second phase. Fifth phase: participants reencountered the 72 animations, each followed by a causality rating. However, the identity-role association was inverted, creating a reversal of associations compared to the third phase. Sixth phase: a manipulation check was conducted.

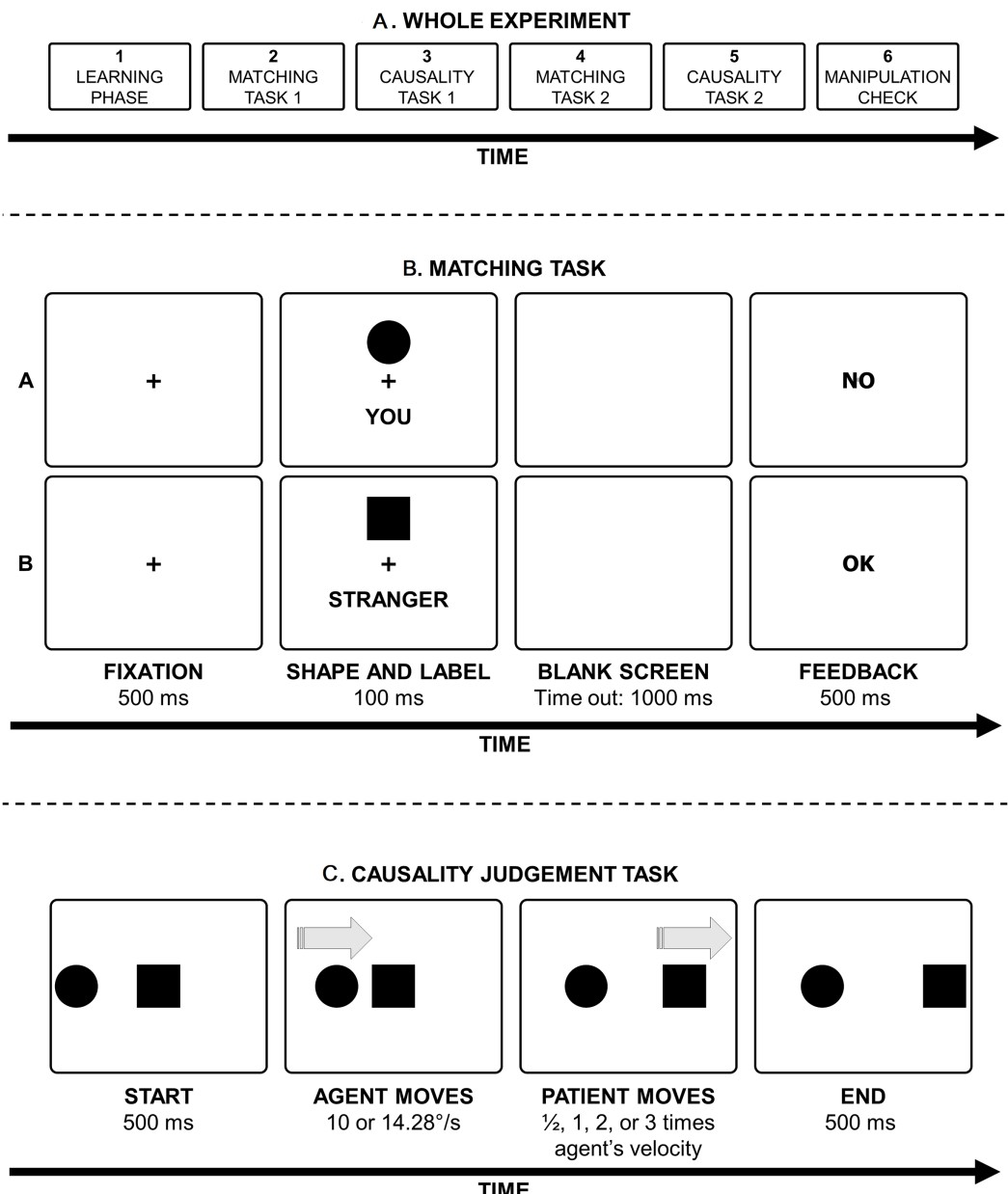

**Figure 2 Sequence and tasks composing the whole experiment.** (A) Illustration of the main tasks that made up the whole experiment. (B) Illustration (not drawn to scale) of the matching task used in Experiments 1 and 2. (A) A trial with the circle shape, the label 'you', and the feedback associated with an incorrect response. (B) A trial with the square shape, the label 'stranger', and the feedback associated with a correct response. The participants decided, by means of a keypress, whether the shape-label association was identical to the association presented in the learning phase. (C) Illustration (not drawn to scale) of the causality judgment task used in Experiments 1. Note that at the end of the animation, participants were presented with the question aimed at assessing the judgement of causality.

In the first phase (*i.e.*, the learning phase), participants learnt the association between two geometrical shapes (a circle and a square) and two identities (the self and a stranger). Specifically, participants were exposed to a sentence on the screen for 40 s, indicating the association between identity and shape. The identity-shape association was counterbalanced across participants. Therefore, each participant encountered either the

sentence 'In this experiment, you are a circle, and a stranger is a square' or the sentence 'In this experiment, you are a square, and a stranger is a circle'. The two shapes were not visually presented during this phase.

In the second phase, participants were asked to complete a matching task (see Fig. 2B), similar to that used by *Sui, He & Humphreys (2012)*. The objective of this matching task was twofold: to strengthen the previously acquired shape-identity association and to test its accurate learning and retention. This task started with a central fixation cross (width: 0.5° of visual angle) for 500 ms (see Fig. 2B). Then one of the two shapes (circle or square) appeared 4° above the fixation (calculated from the centre of the fixation cross to the centre of the shape). The diameter of the circle was 2.3° and the side of the square was 2°; therefore, the two shapes had the same area. At the same time, one of the two words ('tu' or 'sconosciuto', meaning 'you' and 'stranger' in Italian, respectively; 1°/7° width × 0.7° height) appeared 3° below the fixation (calculated from the centre of the fixation cross to the centre of the word). Each shape and word appeared simultaneously on the screen for a duration of 100 milliseconds, followed by the display turning blank. Subsequently, participants were required to indicate—through the act of pressing either the 'A' or 'L' key, with the assignment of keys being counterbalanced among participants—if the presented pair of shape and word corresponded to the association they had previously learned.

The instructions emphasised the importance of responding both quickly and precisely. Depending on whether a participant's response was accurate, inaccurate, or not provided within the 1,000 millisecond timeframe (resulting in a timeout), a visual feedback indicating 'ok', 'no', or 'too slow' would then be shown at the centre of the screen for 500 ms. Each of the four associations of shapes (2: circle *vs*. square) and words (2: you *vs*. stranger) was presented in random order three times in practise and 30 times in the experimental block. Therefore, there were a total of 12 trials in the practise block and 120 trials in the experimental block. We anticipated replicating the findings documented by *Sui, He & Humphreys (2012)* and subsequent studies (*e.g.*, *Frings & Wentura, 2014*; *Dalmaso, Castelli & Galfano, 2019*; *Martínez-Pérez et al., 2024*), namely, quicker and more accurate responses when the word 'you' was correctly linked with the self-related shape compared to all other word-shape combinations. This pattern of results is construed as indicative of the self-prioritisation effect.

The third phase encompassed the main task, the causality judgment task (see Fig. 2C; for a video of a possible trial, see also the Data Availability section). During this phase, participants were randomly presented with the 72 animations described in detail below (plus six practice trials) and provided a causality rating after viewing each animation. In each animation, both shapes (circle and square) were displayed on the screen, horizontally aligned with their centres (Fig. 2C). The patient, one of the shapes, occupied the centre of the screen, while the agent, the other shape, was positioned 12.128° to the left of the centre of the screen (with the patient's leftmost point precisely 10° to the right of the agent's rightmost point). The association between the role (*i.e.*, agent or patient) and the self was counterbalanced among participants. Specifically, for half of the participants, the agent corresponded to the self-related shape, and the patient corresponded to the stranger-related shape (self-agent/stranger-patient association). In contrast, this

association was reversed for the other half of the participants (stranger-agent/self-patient association). Subsequently, 500 ms after the appearance of the agent and the patient, the agent initiated horizontal movement from left to right at a uniform speed towards the patient, halting after traversing 10°. The agent's speed could be either 10 or 14.28°/s. Following a variable delay of −167, 33, or 233 ms from the agent's arrival at its final position, the patient commenced uniform movement in the same direction as the agent, with a speed that could be 0.5, 1, 2, or 3 times as large as the agent's speed. The patient ceased movement after covering 10°. Both the agent and the patient vanished 500 ms after the patient stopped moving.

It is noteworthy that, based on the results of previous studies, social causality impressions are anticipated to be favoured by a negative delay combined with a patient's speed larger than the agent's speed; physical causality impressions by a small positive delay (33 ms) and a patient's speed smaller or equal to the agent's speed; and no clear causality impressions by a long 233 ms delay, irrespective of speed ratios. We do not formulate specific hypotheses concerning the effects of the agent's speed on impressions of social or physical causality. The choice of two different speed levels was primarily made to enhance stimulus variety and alleviate potential monotony in the experimental stimuli. In the experimental block, each participant was randomly presented with a total of 72 stimuli, resulting from a 2 (Agent's speed) × 3 (Delay) × 4 (Agent/Patient speed ratio) × 3 (Repetitions) factorial design. The experimental block was preceded by the practice block, in which participants were presented with six randomly chosen stimuli. It is important to underline that objects always moved from left to right, aligning with the prevalent convention in causal perception experiments. Notably, prior research suggests that the choice of motion direction (left-to-right or right-to-left) does not exert any discernible impact on the resultant impression of causality (*White & Milne, 1997*; *Scholl & Nakayama, 2002*).

After each animation, participants were presented with a question written in the upper part of the screen. Participants tested with the self-agent/stranger-patient association were presented with the question 'How much did YOU cause the motion of the STRANGER?'; participants tested with the stranger-agent/self-patient association were instead presented with the question 'How much did the STRANGER cause YOUR motion?' Participants had to respond using a graphical response scale placed below the written question. The scale consisted of a 17° × 0.1° horizontal bar delimited to the left and right by two short 0.2° × 1° vertical bars, with the labels 'Not at all' and 'Completely' displayed right above the extremes of the scale. Participants were instructed to respond with a mouse click after placing the cursor at the desired point on the horizontal bar. Upon response, the horizontal coordinate of the cursor was recorded and rescaled on a continuum ranging from 0 ('Not at all') to 100 ('Completely').

The fourth and fifth phases were a repetition of the matching and causality judgment tasks, respectively. Everything was identical to the previous matching and causality judgment tasks, with the only two exceptions that trials appeared in a different order, given that they were presented randomly, and in the new causality judgment task (*i.e.*, the fifth phase), the role played by the two shapes was inverted. Specifically, participants who had

been tested in the third phase with the self-agent/stranger-patient association were now tested with the stranger-agent/self-patient association, and *vice versa*. As in the previous causality judgment task, the question presented after each stimulus varied with the role-shape association.

Finally, the sixth phase was a manipulation check task, aimed at assessing whether participants correctly retained in memory the shape-identity association for the whole duration of the experiment. Participants were asked to indicate, by keypress, the accurate description of the association presented at the beginning of the experiment (*i.e.*, first phase). More precisely, one sentence stated that 'In this experiment I was a circle and a stranger a square', while in the other sentence the shape/identity association was inverted. No time limits were provided in this final phase.

# RESULTS

## Data handling

Five participants were removed from the analyses for the following reasons: Three participants provided a wrong response to the final manipulation check task, thus indicating that they did not correctly maintain the association between shape and identity for the whole duration of the experiment; Data from one participant were only partially recorded due to technical failure; One participant exhibited an excessive number of anticipated responses (82.8%) in the matching task, specifically responses with a reaction time (RT) lower than 200 ms (sample mean = 8.2%). It is crucial to acknowledge that in the analysis of data from the matching task, anticipated responses are typically excluded. Therefore, an exceptionally high number of such responses results in a limited dataset for analysis, justifying the exclusion of the data from the mentioned participant. Therefore, the final sample comprised 59 participants (Mean age = 24 years, SD = 4.17, 37 females).

As our primary objective was to examine whether the identity-role association (self-agent/patient-stranger or stranger-agent/self-patient) influences causality judgments, we provide a comprehensive presentation of the results observed in the causality judgment task. For the sake of brevity, the results observed in the matching task are reported here only at the descriptive level, and a detailed description can be found in the dedicated Supplemental File.

## Matching task

As expected, the results replicate those reported by *Sui, He & Humphreys (2012)* and subsequent studies (*e.g.*, *Frings & Wentura, 2014*; *Dalmaso, Castelli & Galfano, 2019*; *Martínez-Pérez et al., 2024*), as we observed quicker and more accurate responses when the word 'you' was correctly linked with self-related shapes compared to all other word-shape combinations. Therefore, the results of the matching task confirmed the presence of a robust self-prioritisation effect.

## Causality judgment task

Participants' responses were initially averaged across the three repetitions of each stimulus. To mitigate interindividual differences arising from subjective interpretations of the rating

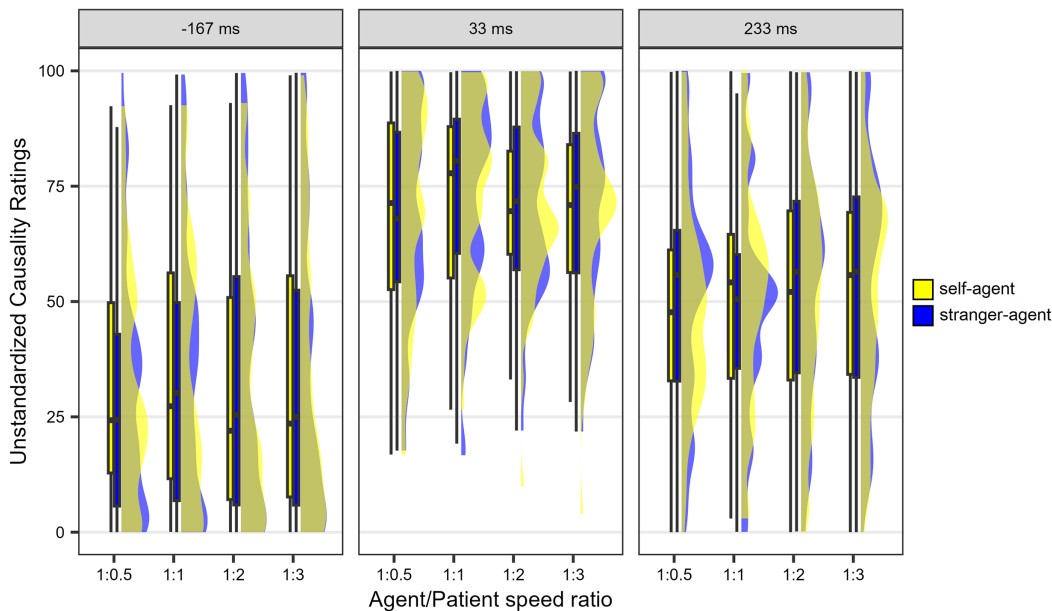

**Figure 3 Results observed in the causality task of Experiment 1.** The animations are arranged on the horizontal axis based on the agent/patient speed ratio, in descending order. The lower and upper hinges of the boxplots correspond to the first and third quartiles of the distribution, the thick horizontal line corresponds to the median. The lower and upper whiskers extend from the lower and upper hinges to the lowest or largest value no further than 1.5 * IQR from the hinge, where IQR is the inter-quartile range. The density curves represent the distribution of the data.

scale, the mean responses of each participant were then standardised using individual means and standard deviations, which were computed based on the complete set of responses provided by the participant in the two causality tasks corresponding to the two possible identity-role associations.

Standardised responses were then analysed by repeated measures four-way analysis of variance (ANOVA) with the following within-participant factors: Identity-role association (2: self-agent/stranger-patient or stranger-agent/self-patient), Agent's speed (2: 10 °/s or 14.28 °/s), Delay (3: −167, 33, or 233 ms), Speed ratio agent/patient (4: 1:0.5, 1:1, 1:2, or 1:3). The Huynh-Feldt method was employed to adjust the degrees of freedom and *p*-values to account for deviations from the assumption of sphericity.

While the analyses were conducted using standardised ratings, unstandardised ratings are presented in both the text and Fig. 3 for the sake of clarity. Additionally, it is important to note that the results in Fig. 3 are averaged across the two levels of agent's speed. This simplification is attributed to the absence of *a priori* hypotheses regarding potential effects of this variable and the observed minimal impact (as detailed below).

The ANOVA results showed that the main effect of Delay was statistically significant ($F$ (1.71, 99.3) = 68.0, $p < 0.001$, $\eta^2_G = 0.346$). *Post-hoc* two-tailed *t*-tests with Bonferroni correction showed that the 33 ms delay was associated with higher causality ratings ($M = 71.02$, $SE = 2.10$) compared with the −167 ms delay ($M = 32.73$, $SE = 3.44$; $t$ (58) = 10.53, $p < 0.001$, $d = 1.37$) and the 233 ms delay ($M = 51.13$, $SE = 2.84$; $t$(58) = 8.45, $p < 0.001$, $d = 1.1$). Moreover, the 233 ms delay was associated with significantly higher ratings than the −167 ms delay, $t$(58) = 4.85, $p < 0.001$, $d = 0.63$. The main effect of Agent's

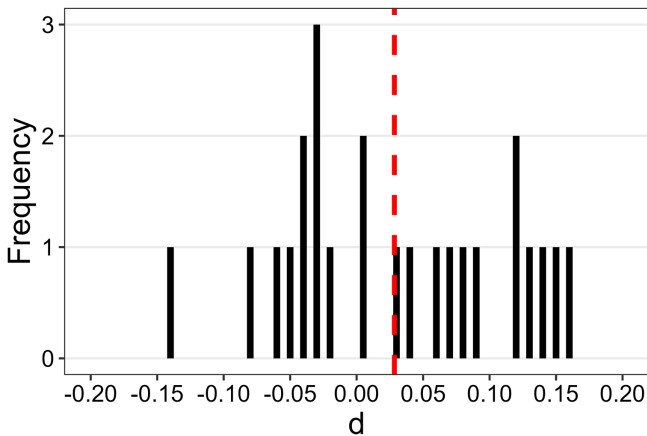

**Figure 4 Distribution of Cohen's *d* for the 24 animations of Experiment 1.** The vertical red dashed line represents the mean *d* (*i.e.*, 0.0284).

speed was also statistically significant ($F(1, 58) = 8.5$, $p = 0.0016$, $\eta^2_G = 0.002$), due to slightly higher causality ratings for the 14.28 °/s speed ($M = 52.22$, $SE = 1.98$) than for the 10 °/s speed ($M = 51.03$, $SE = 1.94$). The main effect of Speed ratio was not statistically significant ($F(1.16, 67.16) = 2.61$, $p = 0.106$, $\eta^2_G = 0.008$). Importantly, the main effect of the Identity-role association was not statistically significant ($F(1, 58) = 0.16$, $p = 0.686$, $\eta^2_G < 0.001$). There was also a significant Delay × Speed ratio interaction ($F(3.73, 216.11) = 2.5$, $p = 0.048$, $\eta^2_G = 0.003$), and *post-hoc* two-tailed *t*-tests with Bonferroni correction showed that, both for the −167 ms and the 33 ms delay, an agent/patient speed ratio equal to one was associated with slightly larger causality ratings than an agent/patient speed ratio equal to 0.5 ($t(58) = 2.83$, $p = 0.03$, $d = 0.37$, and $t(58) = 3.72$, $p = 0.003$, $d = 0.48$, respectively). No other comparison was statistically significant ($ts < 2.3$, $ps > 0.23$). Furthermore, no other interaction was statistically significant ($Fs < 2.5$, $ps > 0.096$).

Importantly, the ANOVA results did not reveal any significant effect of the identity-role association. Because our power analysis was based on an unidirectional hypothesis, we also performed a one-tailed paired sample *t*-test, which confirmed that the causality ratings for the self-agent/stranger-patient association ($M = 51.31$, $SE = 2.04$) were not significantly larger than those for the stranger-agent/self-patient association ($M = 51.93$, $SE = 2.02$; $t(58) = 0.41$, $p = 0.343$, $d = 0.05$). We also performed a Bayesian paired-sample *t*-test using the *ttestBF* function within the *BayesFactor* package (*Morey & Rouder, 2022*). As the prior for H1 we used a half-Cauchy prior distribution with a standard width parameter of 0.707, whereas as the prior for H0 we used a point-null hypothesis. The results showed that the null hypothesis was more than six times as likely as the alternative hypothesis (JZS $BF_{01} = 6.49 \pm 0.08\%$), further supporting the conclusion that the causality ratings were not affected by the identity-role association.

One could speculate that, although the identity-role association does not exhibit a noticeable impact at a global level, it might still influence causality ratings for specific animations. To explore this hypothesis, for each of the 24 animations, we calculated a Cohen's *d* to quantify the effect size of the identity-role association. A positive *d* signifies

that standardised ratings for the self-agent/stranger-patient association were greater than those for the opposite association. Figure 4 displays the distribution of $d$s, revealing that they fell within the range of (−0.2, 0.2). This pattern confirms that none of the animations was associated with a reliable effect of identity-role association.

## DISCUSSION

In our discussion, we first address the unexpected effects of kinematic variables, especially the time delay, on causality judgments.

The results highlight a significant impact of the time delay between the agent's arrival and the patient's departure on the causality ratings (Fig. 3). As expected, a brief 33 ms delay, designed to favour visual impressions of physical causality, resulted in high causality ratings. Surprisingly, a −167 ms delay, intended to favour visual impressions of social causality, resulted in remarkably low causality ratings, even lower than those observed for the long positive delay of 233 ms. Notably, the latter extended delay, traditionally disruptive to visual impressions of causality, led to moderate causality ratings. Therefore, the surprise stems from both the low ratings for animations meant to elicit social causality impressions and the moderate ratings for animations intended to evoke no clear causality impression.

In interpreting these unexpected results, one plausible explanation is that participants implicitly or explicitly interpreted causality in physicalist terms. Contrary to expectations, they may have construed 'causality' as 'physical causality,' resulting in very low causality ratings for animations conveying impressions of 'chasing' or 'intentional reaction.' This interpretation finds support in the study by *Bechlivanidis, Schlottmann & Lagnado (2019)*, where exposure to classic launching animations altered participants' response criteria, indicating a shift in the interpretation of causality towards the concept conveyed by that animation type (*i.e.*, physical causality).

Returning to our experiment results, participants' judgments may have been polarised towards physical causality by animations with clear visual impressions of it (*e.g.*, those with a short 33 ms delay). The absence of contact, a key cue for physical causality, in animations with a negative delay may have led to the interpretation of a non-causal interaction, resulting in very low causality ratings. Animations with a long positive delay, still featuring physical contact between the agent and the patient, could explain their moderate causality ratings. Although the impression of physical causality in these animations was weaker than in those with a short positive delay, it remained stronger than in animations with a negative delay. This hypothesis also accounts for the negligible impact of the speed ratio on participants' ratings. In the context of assessing the presence of physical causality, the presence or absence of physical contact outweighs the speed ratio (from a physical perspective, the agent-patient speed ratio varies widely, depending on object masses). Thus, while speed ratio could influence causality ratings in certain contexts, its impact in our specific scenario was subdued by the more significant cue of physical contact. This further underscores the interplay between low-level visual cues and higher-level evaluation

processes in shaping explicit causality judgments (*Powesland, 1959*; *Brown & Miles, 1969*; *Schlottmann et al., 2006*; *Badler, Lefèvre & Missal, 2012*; *Bechlivanidis, Schlottmann & Lagnado, 2019*; *Deodato & Melcher, 2022*).

Regarding the main research question, the results of the first experiment reveal the absence of an effect from the identity-role association on causality ratings, even though the participants were able to create a strong arbitrary association with the self and a geometrical shape, as anticipated. In contrast to our initial hypothesis, animations associating the self with the agent and a stranger with the patient did not elicit higher causality judgments compared to animations with the reverse association. It is essential to note that because the results suggest that participants evaluated physical causality and not social causality, the null effect of the identity-role association specifically pertains to physical causality. The results of Experiment 1 remain inconclusive regarding the possible effects of the identity-role association on judgments of social causality, which will therefore be the primary focus of Experiment 2.

## Experiment 2

The results of Experiment 1 suggest that animations with distinct visual cues of physical causality tend to bias explicit judgments towards physical causality. In Experiment 2, our objective was to investigate whether the identity-role association influences the judgments of social causality. Therefore, animations associated with visual impressions of physical causality were excluded from the stimulus set. The animations in Experiment 2 were designed to evoke visual impressions of social causality, with a specific focus on the agent 'chasing' the patient.

As a methodological note, it is worth emphasizing that previous studies exploring the perception of social causality can be arbitrarily classified into two categories: (a) studies that used simple but carefully controlled stimuli moving exclusively along the horizontal dimension (*Kanizsa & Vicario, 1968*; *Schlottmann et al., 2006*), enabling precise determination of kinematic parameters favouring social causality impressions in such straightforward situations; (b) studies that employed relatively complex animations featuring multiple speed and/or motion direction changes (*Bassili, 1976*; *Blythe, Miller & Todd, 1999*; *Falmier & Young, 2008*), establishing general principles governing the perception of social causality but making it challenging to isolate specific kinematic parameters. Our current study falls between these two groups, aiming to explore the perception of social causality in the context of bi-dimensional motion, akin to the studies of group (b), while keeping our animations concise and simple to facilitate the exploration of the role of simple kinematic variables, such as the agent's and the patient's relative motion directions, aligning with the studies of group (a). It is worth mentioning that a third group of studies has utilised complex stimuli featuring multiple objects in the scene (for a review, see *Scholl & Gao, 2013*). However, these studies are less relevant to our specific purpose, as they primarily focused on the perceived orientation of stimuli in the scene.

## MATERIALS AND METHODS

### Sample size determination

The theoretical considerations guiding our sample size determination for Experiment 1 were equally applicable to Experiment 2. Therefore, we opted to test 64 participants in Experiment 2 as well.

### Participants

A new sample of 64 naïve students (*Mean age* = 23 years, *SD* = 4.69 years, 48 females), with normal or corrected-to-normal vision, participated in the experiment in exchange for course credits. Before the experiment, they provided a written informed consent. The study was approved by the Ethics Committee for Psychological Research at the University of Padova (approval number: 3455) and was in accordance with the ethical standards of the 1964 Helsinki Declaration.

### Stimuli, apparatus, and procedure

Everything in Experiment 2 mirrored Experiment 1, with participants going through all phases outlined in Fig. 2A. However, specific modifications were made to the causality judgment task, directing participants to focus explicitly on social causality (refer to the Data Availability section for experiment code and a video example). The emphasis was on animations capable of eliciting visual impressions of chasing, achieved by initially reducing the distance between the agent and the patient, followed by maintaining a constant distance after a motion direction change in the agent and/or the patient. This approach aimed to convey the impression that the agent intended to 'catch' the patient, who, in turn, acted to avoid being 'caught.' Further details on the combination of different motion directions and changes are described below. Beyond exploring the main research question ('does the identity-role association modulate social causality judgments?'), our approach sought insights into the critical kinematic parameters for optimal visual impressions of social causality in such simple stimulus situations.

Twelve distinct animations (coded A–L in Fig. 5 and Table 1) were created. Figure 5 offers a schematic representation of each animation, while Table 1 provides technical information regarding the spatial coordinates of the agents' and patients' motion direction changes. Each animation can be arbitrarily divided into three steps. The first step, lasting 2,300 ms, started with the agent and the patient remaining stationary for 1,000 ms. Afterward, the agent and patient performed two brief 'vertical hops on the spot' (each hop lasting 400 ms, with a height of 2.9°) to enhance the visual impression that the shapes were animated and capable of self-propelled motion (*Parovel, 2023*). These two 'hops' are not represented in Fig. 5. Afterward, the agent and the patient remained stationary for an additional 500 ms before the second step commenced.

The second step corresponded to the initiation of motion for at least one of the two objects and is represented in the second panel of each row in Fig. 5. This phase lasted for 700 ms, during which the agent could move either rightward (animations A to F) or downward (animations G to L), and the patient could remain stationary (A, B, G, H), move leftward (C, D, I, J), or move upward (E, F, K, L). Both possible motion directions of the

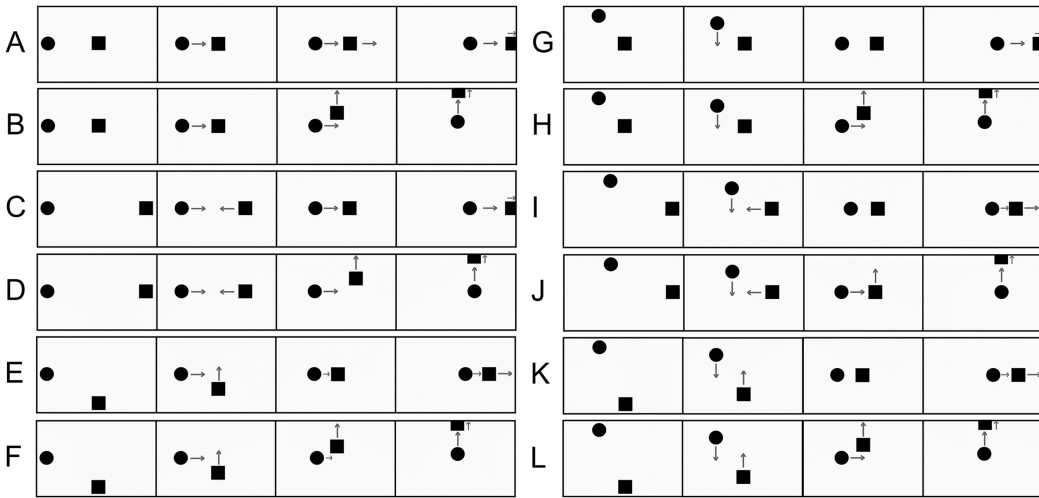

**Figure 5 Schematic representations of the twelve animations used as stimuli in Experiment 2.** The first panel in each row is referred to as the 'first step' of the animation in the main text, the second panel as 'second step' and the third and fourth panel as 'third step'. The stimuli are not drawn to scale. To simplify the main features of the animations, some of their elements have been intentionally left out from this representation, namely the two 'vertical hops on the spot' that the shapes performed during the first step, and the two vertical rectangles that delimited the screen to the left and to the right. At the end of the animation, participants were presented with the question aimed at assessing the judgement of causality. A summary description of the kinematic characteristics of each animation is provided in Table 1. Please note that the layout of the figure may convey the impression that the two moving objects are observed from above a flat horizontal surface, contrasting with the customary lateral perspective seen in launching or intentional reaction animations (see Fig. 1). It is important to note that this perception is likely a result of the layout of the static depiction. The accompanying videos of the stimuli, available on OSf (http://dx.doi.org/10.17605/OSF.IO/E6HD7), indicate a more conventional lateral perspective.

**Table 1 Summary description of the characteristics of the twelve animations used as stimuli in Experiment 2.**

| | FIRST STEP | | SECOND STEP | | THIRD STEP | | | |
|---|---|---|---|---|---|---|---|---|
| Animation | Agent's initial position | Patient's initial position | Agent's motion | Patient's motion | Agent's direction change position | Agent's motion | Agent's direction change position | Patient's motion |
| **A** | (−12.1, 0) | (0, 0) | Rightward | Stationary | / | Rightward | (0, 0) | Rightward |
| **B** | (−12.1, 0) | (0, 0) | Rightward | Stationary | (0, 0) | Upward | (0, 0) | Upward |
| **C** | (−12.1, 0) | (12.1, 0) | Rightward | Leftward | / | Rightward | (2.1, 0) | Rightward |
| **D** | (−12.1, 0) | (12.1, 0) | Rightward | Leftward | (2.1, 0) | Upward | (2.1, 0) | Upward |
| **E** | (−12.1, 0) | (2.1, −10) | Rightward | Upward | / | Rightward | (2.1, 0) | Rightward |
| **F** | (−12.1, 0) | (2.1, −10) | Rightward | Upward | (2.1, 0) | Upward | / | Upward |
| **G** | (−4, 10) | (0, 0) | Downward | Stationary | (−4, 0) | Rightward | (0, 0) | Rightward |
| **H** | (−4, 10) | (0, 0) | Downward | Stationary | (−4, 0) & (0, 0) | Rightward & Upward | (0, 0) | Upward |
| **I** | (−3, 10) | (12.1, 0) | Downward | Leftward | (−3, 0) | Rightward | (2.1, 0) | Rightward |
| **J** | (−4, 10) | (12.1, 0) | Downward | Leftward | (−4, 0) & (2.1, 0) | Rightward & Upward | (2.1, 0) | Upward |

(Continued)

| | FIRST STEP | | SECOND STEP | | THIRD STEP | | | |
|---|---|---|---|---|---|---|---|---|
| Animation | Agent's initial position | Patient's initial position | Agent's motion | Patient's motion | Agent's direction change position | Agent's motion | Agent's direction change position | Patient's motion |
| K | (−4, 10) | (0, −10) | Downward | Upward | (−4, 0) | Rightward | (0, 0) | Rightward |
| L | (−4, 10) | (0, −10) | Downward | Upward | (−4, 0) & (0, 0) | Rightward & Upward | / | Upward |

**Note:**
The spatial coordinates $(x, y)$ of initial positions and motion direction change positions are expressed in degrees of visual angle; note that (0, 0) corresponds to the centre of the screen, and that the left and the bottom halves of the screen are denoted with negative $x$ and $y$ coordinates, respectively. The '/' symbol indicates that no motion direction change occurred.

agent were combined in a factorial manner with the three possible motion directions of the patient. During motion, the speed of both objects was kept fixed at 18°/s, which means that the shapes moved faster with respect to Experiment 1. This was meant to enhance further the visual impression of animated motion (*Parovel, 2023*). It is crucial to note that the duration of the second step was set to make the agent and the patient visually close at the end while preserving a visible spatial gap between them, a crucial feature distinguishing social from physical causality.

The third step began when at least one of the two objects changed its motion direction, represented in the third and fourth panels of each row in Fig. 5. Lasting 1,300 ms, the objects moved at 18°/s, with the agent and the patient always moving in the same direction at the end, either rightward (animations A, C, E, G, I, K) or upward (animations B, D, F, J, L). Combining these two final motion directions with the two initial motion directions of the agent and the three initial motion directions of the patient resulted in the full set of 12 animations used in this experiment. When the agent initially moved downward and the final motion direction of both the agent and the patient was upward (animations H, J, L), the agent performed two motion direction changes. The first change (from downward to rightward) occurred at the beginning of the third step, and the second (from rightward to upward) occurred after 700 ms. In other animations, the agent performed no motion direction change (A, C, D) or just one motion direction change (B, D, F, G, I, K). The patient performed no direction change in animations F and L, and one direction change in all other cases. In sum, the total duration of each animation was 4,300 ms (2,300 ms first step + 700 ms second step + 1,300 ms third step).

Two additional details merit mention. Firstly, each animation was framed on the left and right by two visible black rectangles throughout the entire animation duration (width: 0.3 of normalised units; height: full screen height, corresponding to 2 normalised units). These are not represented in Fig. 5. This framing aimed to create the impression that the shapes moved on a hypothetical 'stage'. Secondly, unlike Experiment 1, the agent and the patient remained in motion when the animation disappeared from the screen. In certain cases, the patient had already vanished beyond the screen's margin by the animation's termination. This deliberate choice aimed to convey the impression that the 'chase' could extend beyond the confines of the 'stage' represented by the screen.

As in Experiment 1, horizontal motions were always from left to right, aligning with the prevalent convention in causal perception experiments.

As in Experiment 1, after each animation, participants responded to the question 'How much did YOU cause the motion of the STRANGER?' in the case of the self-agent/ stranger-patient association, and to the question 'How much did the STRANGER cause YOUR motion?' for the opposite association. Therefore, the questions were the same in both experiments. The response scale details mirrored those in Experiment 1. In each of the two causality tasks, corresponding to the two possible identity-role associations, participants were randomly presented with a total of 36 stimuli. This resulted from a 12 (Animations) × 3 (Repetitions) factorial design. Preceding each experimental block, participants underwent a practice block wherein they were randomly presented with each of the 12 animations.

## RESULTS

### Data handling

Data were treated as in Experiment 1. Two participants were removed from the analyses because they provided a wrong response in the final manipulation check task. Therefore, the final sample consisted of 62 participants (*Mean age* = 23 years, *SD* = 4.76 years, 47 females).

### Matching task

As in Experiment 1, for the sake of brevity, we only report here that the results confirmed the presence of a robust self-prioritisation effect, in line with *Sui, He & Humphreys (2012)* and with the results of Experiment 1. For a detailed description of the results, please see the dedicated Supplemental File.

### Causality judgment task

As in Experiment 1, participants' responses were initially averaged across the three repetitions of each stimulus. Then they were standardised using individual means and standard deviations computed based on the complete set of responses provided by the participant in the two causality tasks.

Standardised responses were then analysed by repeated measures two-way analysis of variance (ANOVA) with the following within-participant factors: Identity-role association (2: self-agent/stranger-patient or stranger-agent/self-patient), Animation type (12: animations A–L, see Fig. 5). The Huynh-Feldt method was employed to adjust the degrees of freedom and *p*-values to account for deviations from the assumption of sphericity. For clarity, unstandardised ratings are presented in both the text and Fig. 6.

Before delving into the ANOVA results, it is important to highlight a significant consideration regarding animation A, representing a typical 'intentional reaction' sequence (*Kanizsa & Vicario, 1968*). Disregarding the identity-role association factor, this animation garnered high mean causality ratings ($M = 61.56$, $SE = 2.43$). Notably, this value starkly contrasts with the low mean causality ratings observed in Experiment 1 for animations that were structurally similar to animation A (*i.e.*, those resulting from the combination of

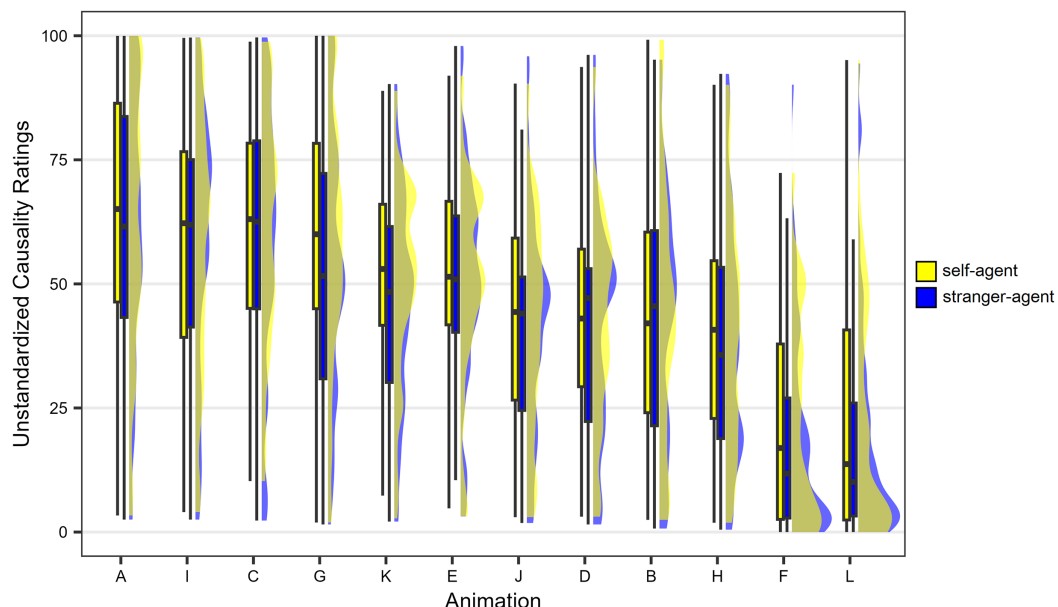

**Figure 6 Results observed in the causality task of Experiment 2.** The animations are arranged on the horizontal axis based on the observed causality ratings, in descending order. The lower and upper hinges of the boxplots correspond to the first and third quartiles of the distribution, the thick horizontal line corresponds to the median. The lower and upper whiskers extend from the lower and upper hinges to the lowest or largest value no further than 1.5 * IQR from the hinge, where IQR is the inter-quartile range. The density curves represent the distribution of the data.

negative delay and a 1:1 speed ratio). A Welch two-sample $t$-test indicated that the unstandardised mean ratings of causality for animation A in Experiment 2 (averaged across two identity-role association levels) were significantly greater than the unstandardised mean ratings of causality for animations in Experiment 1 characterised by negative delay and a 1:1 speed ratio (averaged across agent's speed and identity-role association levels) ($t(117.4) = 6.01$, $p < 0.001$, $d = 1.09$). The substantial difference in mean causality ratings, despite the strong similarities between the two animation types, suggests a distinct response criterion for participants in the two experiments. Specifically, participants in Experiment 2 seemed to focus on evaluating social causality rather than physical causality. The results of this comparison support the hypothesis that by excluding animations depicting physical causality in Experiment 2, we effectively guided participants toward assessing social rather than physical causality.

Moving to the ANOVA results, the main effect of the Identity-role association was not statistically significant ($F(1, 61) = 1.68$, $p = 0.20$, $\eta^2_G = 0.005$), whereas the main effect of Animation type was statistically significant ($F(5.10, 311.34) = 62.18$, $p < 0.001$, $\eta^2_G = 0.372$). The two-way interaction was not statistically significant ($F(8.27, 504.59) = 1.05$, $p = 0.40$, $\eta^2_G = 0.004$). Mean standardised causality ratings for the 12 animations, averaged across the two identity-role associations, were compared using *post-hoc* two-tailed $t$-tests. With a large number of comparisons, *Hochberg*'s *(1988)* sequentially acceptive step-up Bonferroni procedure was applied (see also *Keselman, 1994*), resulting in the identification of four different groups of animations (Table 2). Specifically, standardised causality ratings

**Table 2 Mean unstandardized causality ratings for the animations employed in Experiment 2.**

| Group | Animations | Mean unstandardized causality ratings (with SE) |
|---|---|---|
| 1 | A, C, G, I | $M = 58.67$, $SE = 2.33$ |
| 2 | E, K | $M = 49.24$, $SE = 1.84$ |
| 3 | B, D, H, J | $M = 41.11$, $SE = 2.06$ |
| 4 | F, L | $M = 20.22$, $SE = 1.9$ |

Note:
The animations are grouped according to the outcomes of *post-hoc* comparisons.

for animations A, C, G, and I were not significantly different from each other and were all significantly larger than those for the other animations (except for the comparisons between G and E, I and E). Moreover, standardised causality ratings for animations E and K were not significantly different from each other and were both significantly larger than those for animations B, D, F, H, J, L (except for the comparison between K and B). Lastly, standardised causality ratings for animations B, D, H, J were significantly larger than those for animations F and L. These results will be further discussed in the next section.

As for the main research question, the ANOVA results indicated that the main effect of the identity-role association and its interaction with animation type were not statistically significant. Since the sample size determination was based on a unidirectional alternative hypothesis (with $d = 0.4$), a one-tailed paired sample $t$-test, as in Experiment 1, was performed, confirming that the causality ratings for the self-agent/stranger-patient association ($M = 46.07$, $SE = 1.99$) were not significantly larger than those for the stranger-agent/self-patient association ($M = 43.60$, $SE = 2.03$; $t(61) = 1.29$, $p = 0.10$, $d = 0.16$). A Bayesian paired-sample $t$-test using the *ttestBF* function within the *BayesFactor* package (*Morey & Rouder, 2022*) further supported this conclusion, showing that the null hypothesis was more than three times as likely as the alternative hypothesis (JZS $BF_{01} = 3.26 \pm 0.05\%$). This lends additional support to the finding that the identity-role association did not affect causality ratings. Note that, as in Experiment 1, as the prior for H1 we used a half-Cauchy prior distribution with a standard width parameter of 0.707, whereas as the prior for H0 we used a point-null hypothesis.

To examine whether, despite the absence of an overall effect, the identity-role association had a noticeable impact on some of the 12 animations, we computed 12 Cohen's $d$ values to quantify the effect size of the identity-role association for each animation. A positive $d$ indicates that standardised ratings for the self-agent/stranger-patient association were higher than those for the opposite association. Figure 7 illustrates the distribution of these $d$s. None of them reached the 0.4 threshold that we considered scientifically interesting in this context. However, some differences from the analogous Cohen's $d$s distribution of Experiment 1 (see Fig. 4) are noteworthy. Firstly, one $d$ value ($d = 0.31$ for animation G) is not too distant from the 0.4 threshold. Secondly, all but two $d$ values were greater than zero, indicating that, for most of the animations, the causality ratings for the self-agent/stranger-patient association were slightly higher than those for the opposite association.

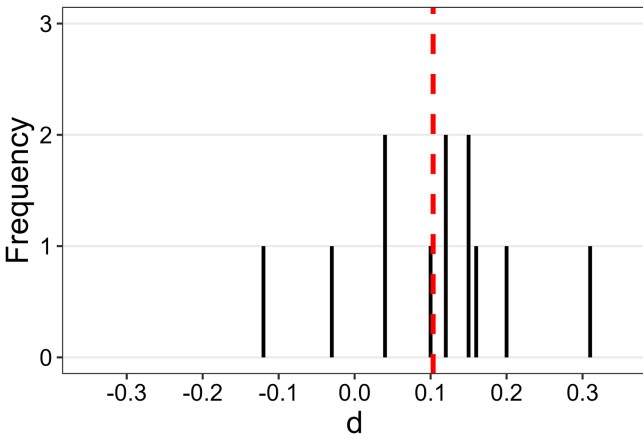

**Figure 7 Distribution of Cohen's *d* for the 12 animations of Experiment 2.** The vertical red dashed line represents the mean *d* (*i.e.*, 0.1033).

## DISCUSSION

By limiting the set of stimuli to animations showing no physical contact between the agent and the patient, we successfully directed participants to focus on social causality rather than physical causality. This is evident from the relatively high causality ratings provided by participants for some animations (see Fig. 6), particularly animation A. It is noteworthy that in Experiment 1, animations sharing all relevant aspects with animation A received very low causality ratings (see Fig. 3). This suggests a different interpretation of the concept of causality by participants in Experiment 2 compared to those in Experiment 1. This aligns with the findings of a prior study by *Bechlivanidis, Schlottmann & Lagnado (2019)*, indicating that the presence of stimuli eliciting the launching effect can shift participants' response criterion towards a more restrictive interpretation of causality. In Experiment 2, by not presenting such launching stimuli, we maintained a response criterion allowing for a more liberal interpretation of causality, including social causality. This underscores the concept that explicit causality judgments are influenced not only by low-level kinematic processes but also by high-level evaluation processes.

In summary, Experiment 2 results indicate that, akin to physical causality ratings, social causality ratings remained largely unaffected by the identity-role association. However, it is important to note some distinctions between the results of this experiment and those of Experiment 1. Firstly, in Experiment 1, the effect size for the identity-role association was negligible (*d* = 0.05), whereas in Experiment 2, a small effect emerged (*d* = 0.16). This suggests that associating the self with the agent and a stranger with the patient led, overall, to slightly higher causality ratings compared to the opposite association. However, this small effect falls short of the minimum effect size that we considered scientifically interesting in the context of explicit causal reports (*i.e.*, *d* = 0.4). Secondly, in Experiment 1, the effect size for the identity-role association never exceeded 0.2 for any animations used, and the distribution of *d*s was symmetrically centred around zero (Fig. 4). Conversely, in Experiment 2, a *d* value of 0.31 for animation G was observed, and the distribution of *d*s was not symmetrical around zero but shifted towards the positive end of the continuum

(Fig. 7). Overall, these comparisons indicate that in Experiment 2 the effects of the identity-role association were not totally negligible as in Experiment 1. However, the observed effects are minimal to the extent that they can be deemed uninteresting from a theoretical standpoint, particularly when juxtaposed with the strong effects of the kinematic parameters of the animations.

The unequivocal finding of Experiment 2 is that kinematic parameters of the animation exerted a significant impact on causality ratings, with the effect size for the Animation factor being over 70 times larger than that of the Identity-role factor. Therefore, it can be concluded that social causality judgments were predominantly influenced by kinematic parameters and minimally affected by the identity-role association. Notably, animations F and L, which received the lowest ratings, were characterised by a lack of change in the patient's motion pattern, as it continued moving upward throughout the entire sequence. This underscores the necessity of a change in the patient's motion pattern for the perception of social causality, aligning with the documented importance of temporal contiguity between the agent's and patient's motions (Bassili, 1976).

Animations B, D, H, and J, featuring the patient's upward motion after a motion direction change, received intermediate causality ratings. Unlike animations with lower ratings, these animations included a motion direction change of the patient. However, unlike animations with higher ratings, both the agent and the patient moved upward after the motion direction change, rather than rightward. A possible explanation for these animations receiving lower ratings, compared to those where both the agent and the patient moved rightward, relates to the screen shape and initial position of the stimuli. In the context of upward motion, there was a shorter distance between the position of the motion direction changes and the moment when the patient crossed the screen border. This potentially shortened the time window during which the agent could be perceived as 'chasing' the patient, moving behind it with the same motion direction and speed. This might have made the chasing impression less compelling compared to animations where both the agent and the patient moved rightward.

Animations E and K received slightly lower causality ratings than animations A, C, G, and I. The exact reason for this discrepancy remains inconclusive. While one might speculate that the social causality impression was more compelling when the agent continuously moved rightward throughout the animation, this hypothesis contradicts the observed high ratings for animations G and I, which featured a 90° motion direction change of the agent. A distinguishing characteristic of E and K is that the patient initially moved upward, unlike animations A and G (stationary patient) or C and I (leftward motion). The initial upward motion of the patient in E and K might have, for some reason, reduced the saliency of the subsequent motion direction change.

In the context of the employed stimuli, two crucial kinematic characteristics influencing the perception of social causality emerged: (1) the presence of the patient's motion direction change and (2) the rightward motion of both the agent and the patient in the final animation step. This likely enhanced the visibility of the 'chase' scenario for a sufficient duration. Notably, other kinematic features seemed to exert minimal or no impact on explicit judgments of social causality. Specifically, relatively high ratings were consistent,

regardless of whether the agent initially moved rightward (animations A, C, E) or downward, forming a 90° angle with the agent's and patient's final directions (G, I, K). Additionally, ratings remained relatively high irrespective of whether the patient was initially stationary (A, G) or moved leftward (C, I), with upward motion associated with slightly lower causality ratings (E, K).

## General discussion

Visual impressions of causality exhibit a marked asymmetry, wherein one object is perceived as the causal agent, and the other as the patient undergoing the agent's causal action. In two experiments, we investigated whether associating the self with the agent and a stranger's identity with the patient would lead to higher ratings of physical or social causality compared to the opposite association. The following discussion presents the main results in increasing order of importance.

First, participants successfully established an arbitrary association between two identities (self and stranger) and two shapes (square and circle). The observed results in the matching tasks replicate the well-known self-prioritisation effect (*Sui, He & Humphreys, 2012*), indicating participants' ability to form and retain arbitrary associations between shapes and identities. Additionally, with few exceptions, participants maintained the shape-identity association throughout the experiment, as verified by the results of the final manipulation check.

Second, in Experiment 1, stimuli expected to evoke visual impressions of social causality received remarkably low ratings on the causality scale. We attributed this outcome to the presence of animations in the stimulus set that induced the launching effect. According to the findings of a prior study by *Bechlivanidis, Schlottmann & Lagnado (2019)*, the launching effect may have shifted participants' response criterion toward physical causality, where the primary cue is the physical contact between the agent and the patient. This shift, which indicates the involvement of high-level evaluation processes in explicit judgments of causality, may have led participants to evaluate the social causality stimuli without physical contact as non-causal. Experiment 2 supported this hypothesis, as excluding stimuli featuring physical causality resulted in higher causality ratings for animations depicting social causality. Overall, these results underscore how participants' interpretation of the concept of causality in an explicit judgment task is significantly influenced by context, particularly the composition of the set of stimuli within the experiment.

Third, in Experiment 2, the manipulation of different motion directions and motion direction changes for both the agent and the patient provided insights into the kinematic features influencing judgments of social causality. The findings indicated that the absence of a patient's motion direction change resulted in low causality judgments. Additionally, when both the agent and the patient moved upward after motion direction changes, causality ratings were moderately low. This was attributed to upward motion shortening the time window during which the agent could be perceived as 'chasing' the patient. In summary, it seems that judgments of social causality were favoured by a relatively long
'chasing' phase, and various combinations of motion directions and changes led to moderate-to-high causality ratings. Notably, the study demonstrated that visual impressions of social causality could emerge even with initial motion directions forming a 90° angle with the agent's and patient's final motion directions, as observed in animations G, E, I, and K (Fig. 5). Overall, the presence of a 'chasing' sequence in the second part of the animation outweighed the importance of preceding events in influencing causality judgments.

It is important to note that the results of Experiment 2 offer a limited perspective on the cues influencing visual impressions of social causality. First, the manipulation of kinematic parameters focused solely on motion directions and did not consider other variables known to potentially impact social causality perception, such as the agent's and patient's speed ratio, and the spatial gap between objects (Kanizsa & Vicario, 1968). Second, the agent's and patient's motions were constrained to horizontal and vertical straight trajectories, excluding oblique trajectories. These constraints naturally limit the generalizability of conclusions regarding the influence of kinematic parameters on social causality judgments. Future studies should explore how various motion directions and changes, including non-horizontal and non-vertical dimensions, interact with other kinematic parameters, such as speed ratio and spatial gap. However, it is crucial to recognize that while these limitations are important, the primary goal of manipulating kinematic parameters was to create a diverse set of stimuli for investigating the impact of identity-role associations on explicit causality judgments.

Fourth, the most important finding pertains to the impact of the identity-role association on explicit judgments of physical and social causality. To recapitulate the pertinent background, our main hypothesis was grounded in two distinct lines of evidence from the literature on causal perception and cognition. On the one hand, several studies suggest that explicit judgments of physical and social causality result from an interplay between genuine visual processes and high-level causal reasoning (e.g., Heider & Simmel, 1944; Shor, 1957; Thayer & Schiff, 1969; Badler, Lefèvre & Missal, 2012; Vicovaro, 2018; Bechlivanidis, Schlottmann & Lagnado, 2019; Deodato & Melcher, 2022). On the other hand, literature on causal cognition indicates a strong tendency for individuals to perceive themselves as causal agents, leading them to underestimate the role of contextual factors in the visible outcomes of physical and social interactions (e.g., Langer, 1975; Thompson, 1999; White, 2006b, 2007). Drawing from these insights, we hypothesised that the inclination to perceive the self as a causal agent could interact with kinematic variables, resulting in a decrease in causality ratings when the participant identified herself as the patient and a stranger as the agent, compared to the opposite scenario. However, the results of Experiment 1 unequivocally refute this hypothesis. They indicate that there are definite limits to the influence of non-kinematic variables on judgments of visual stimuli representing physical causality. In simpler terms, explicit causal judgments of visual displays remain impervious to the 'self as a causal agent' bias and are driven instead by low-level kinematic cues. However, it is important to note that this does not imply that explicit causality judgments are solely determined by kinematic cues. The distinct

interpretation of 'causality' between the two experiments suggests that higher-level semantic processes interact with the lower-level processes based on kinematic cues, shaping explicit judgments.

In Experiment 2, the rejection of the alternative hypothesis concerning the impact of the identity-role association was not as definitive as in Experiment 1. Nevertheless, the observed effect was of such minimal magnitude that it holds no theoretical significance, especially when contrasted with the considerably larger effect of animation type, surpassing it by more than 70 times. The predominant observation is that the impact of identity-role association on explicit judgments of social causality remains subtle, bordering on negligible.

A potential limitation of our study is the possibility that the shape-identity associations established through our experimental procedure, while eliciting a reliable self-prioritisation effect in the matching task, might not have been sufficiently deep and robust. Participants may not have genuinely identified themselves with the agent or the patient. Future research could explore alternative shape-identity manipulations, such as associating the self with a stimulus with social content, like an avatar, and/or allowing participants to directly control the motion of the self-associated stimulus on the screen using keys before undertaking the causality judgment task. The former manipulation may establish a more direct and experiential association between the self and the corresponding shape, while the latter may convey the perception that the self-associated shape possesses voluntary and self-initiated motion, enhancing participant' perception of its autonomy, even when assuming the role of the patient in a causal interaction.

## CONCLUSIONS

The hypothesised scenario was that, within the context of simple animations showing physical or social causal interactions between an agent and a patient, establishing an association where the self was linked with the patient and a stranger with the agent could result in diminished causality ratings in comparison to the reverse association.
The underlying logic behind this hypothesis is that the perception of the self as a potential causal agent could extend to the shape affiliated with the self, thereby diminishing the impression that its motion was solely due by the causal action of the agent. We explored this hypothesis using a novel experimental task inspired by the literature on the self-prioritisation effect. The results of both experiments contradict the initial hypothesis. Overall, the outcomes support the idea that the self-concept as a causal agent does not significantly affect causal judgments in simple visual displays, where kinematic cues play a predominant role. The present study sought to bridge the gap between research on causal perception using simple visual displays and studies on causal reasoning involving verbal descriptions and abstract stimuli. The key findings highlight that explicit judgments of physical and social causality in simple animations are not significantly impacted by cognitive bias where individuals tend to overestimate the self's contribution to observed outcomes in a given context.

## ACKNOWLEDGEMENTS

We thank PeerJ for their support in our work through the #OAadvocates 2022 initiative.

### Funding

This work was supported by the "Ministero dell'Università e della Ricerca"-MUR (PRIN 2022 PNRR P2022TPX8E). The funders had no role in study design, data collection and analysis, decision to publish, or preparation of the manuscript.

### Grant Disclosures

The following grant information was disclosed by the authors:
"Ministero dell'Università e della Ricerca"-MUR: PRIN 2022 PNRR P2022TPX8E.

### Competing Interests

Mario Dalmaso is an Academic Editor for PeerJ.

### Author Contributions

- Michele Vicovaro conceived and designed the experiments, analyzed the data, prepared figures and/or tables, authored or reviewed drafts of the article, and approved the final draft.
- Francesca Squadrelli Saraceno performed the experiments, authored or reviewed drafts of the article, and approved the final draft.
- Mario Dalmaso conceived and designed the experiments, analyzed the data, prepared figures and/or tables, authored or reviewed drafts of the article, and approved the final draft.

### Human Ethics

The following information was supplied relating to ethical approvals (*i.e.*, approving body and any reference numbers):

Ethics Committee for Psychological Research at the University of Padova (approval number: 3,455).

### Data Availability

The raw data, video examples, and experiment code are available at Open Science Framework: Vicovaro, Michele, and Mario Dalmaso. 2024. "Exploring the Influence of Self-Identification on Perceptual Judgments of Physical and Social Causality." OSF. February 13 http://dx.doi.org/10.17605/OSF.IO/E6HD7.

### Supplemental Information

Supplemental information for this article can be found online at http://dx.doi.org/10.7717/peerj.17449#supplemental-information.

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
