# Peer review of "Exploring the influence of self-identification on perceptual judgments of physical and social causality"

_PeerJ, doi:10.7717/peerj.17449_

## Round 0.1 · original submission · Major Revisions

I have read the manuscript myself, as well as the reviewers' comments: I agree that it has merit, but at the same time there is room for improvement. To this purpose, I invite you to carefully adress all the comments below, in order to improve the overall quality and clarity of the manuscript.

·

Basic reporting

This is a very clear and well-written paper, and the literature review is both comprehensive and appropriate. I particularly appreciate the inclusion of some Italian work I was previously unaware of that I hope to learn more about in the near future.

I do have one comment about the figures: In figs. 4 and 6 it would be helpful to reduce the opacity of the self-agent density distributions by about 50% to make both distributions (and their overlap) more visible.

Experimental design

The methods are clear and the design makes sense, and I have no technical or ethical concerns.

Validity of the findings

The conclusions about Experiment 2 go a little beyond the data, in my opinion. There really is no effect of identity, at least not a reliable one. Yes, there is a hint of an effect for a few animations, but to draw the conclusion that there might be a meaningful result there, I would want to see an additional experiment with just those animations that showed a clearer result. I would remove some of the language in the general discussion about there being hints of an effect, or run the additional experiment to establish the result more definitively one way or the other.

Additional comments

I have a general comment about the self/other manipulation which I think is just a limitation that should be discussed. First, while the authors do successfully replicate the self-identification RT advantage, I think that this is overall not a very strong identity manipulation, and the RT advantage, while reliable, is not necessarily evidence of true self-identification. I would be interested to see future projects where participants actually got experience controlling one of the shapes as an avatar as part of the manipulation. However, for the purposes of the current paper, I think it is sufficient to discuss the possibility that the identity manipulation itself wasn't strong enough, and that a stronger manipulation (and one that tied identity more to kinematics) might show a different pattern.

I also found the discussion of the use of explicit judgments in the introduction very interesting, but I felt it was somewhat undercut by the point made in Experiment 2 that the interpretation of "causality" seems to be different in each experiment due to the inclusion or absence of the launching event. It's unclear what "everyday causal judgments" actually are or if these judgments can capture them, if the interpretation of the whole construct a causal judgment scale is meant to measure is affected by the inclusion of launching. I think that it might be worth revising this part of the introduction slightly to mention this issue, bringing in the Bechlivanidis et al. (2019) work to contextualize the scales used throughout the paper.

Reviewer 2 ·

Basic reporting

Non-kinematic Information. The idea that “specific nature of non-kinematic information that can be integrated with visual impressions of causality remains largely unknown” (lines 190-192) might be an overstatement. For example, “force” per se is not a visual property (although the consequences of force can be visual) and White, as well as Hubbard and Ruppel, have published regarding the perception of force and how that relates to the perception of causality. Gilden and Proffitt (Gilden, 1991; Gilden & Proffitt, 1989; Proffitt & Gilden, 1989) have also addressed use of kinematic information (and derivation of non-kinematic information) in displays that depicted colliding stimuli.

Social Attribution. I wonder how much judgments of social causality might be influenced by biases in social attributions. For example, in the current manuscript, participants are in essence judging how much they or a stranger influence each other in the form of the visual display, and this seems potentially related to the egocentric bias in social psychology (i.e., when considering the joint outcome of a project, we tend to overestimate our contributions and underestimates the contributions of another perron). Perhaps some discussion of how social attribution biases might influence judgments of social psychology are in order.

Lines 46-47: The launching effect (as well as other demonstrations of physical casualty and social causality) were extensively reviewed in Hubbard (2013a,b). Although perhaps a bit dated now, it might nonetheless be helpful to readers to cite those reviews here and elsewhere in the manuscript as appropriate.

Lines 88-92: This mixes discussion of delay with discussion of B’s speed. It might be easier for readers if only one variable were dealt with at a time (e.g., discuss all aspects of delay before then discussing aspects of B’s speed).

Lines 127-137: Perhaps which features are considered as “low-level process” and which features are considered as “high-level inferences” should be justified? Movement might be a low-level feature, but then wouldn’t interpretation of the pattern of movement be a high level feature? The authors suggest “shape, depth, and mechanical causality” as low-level, but if that were correct, they wouldn’t seem to be equally low-level (e.g., if causality in movement is perceived, that would seem to imply a more basic perception of an object [i.e., shape]).

Line 162: The work of Gilden and Proffitt seems relevant here (see also General Issue D).

Lines 216-222: In addition to the causal asymmetry discussed by White (2006, 2007), White (2012) and Hubbard and Ruppel (2013, 2017) discussed a force asymmetry (the force attributed to A and exerted on B is greater than the force attributed by B exerted on A), and this is consistent with the notion of non-kinematic information being integrated with visual perception of causality noted earlier.

Lines 260-262. The study by Bloom and Veres (1999) regarding groups rather than individuals seems relevant, as it also involves identification with geometric stimuli, albeit of larger groups. Rather than the individual participant.

Line 605: The use of the word “phases” to describe both the three steps of each animation, as well as the (apparently) six steps of each trial (see General Issue B), could be confusing to readers. I suggest one of the uses be changed to “steps” to some other term.

Lines 640-644: The statement “the agent and patient remained in motion when the animation disappeared from the screen… this deliberate choice aimed to convey the impression that participants observed a fragment of a more extensive changing sequence” was confusing to me. How could participants see the agent or patient after those stimuli disappeared from the screen? Wouldn’t the act of disappearing prevent such a perception? And if the agent and patient couldn’t be seen because they'd disappeared, then how would that convey that the appearance the participants had seen was a fragment of a longer chase?

Lines 834-835: But didn’t phase 4 or 5 within each trial imply a change in this association within each trial?

Line 842: “evaluate” implies an inference or decision making process and would not suggest, at least to me, a direct perception.

Lines 863-864: “and did not consider other variables known to potentially impact social causality perception, such as the agent’s and patient’s speed ratio…” I don’t understand this. Agent/patient speed ratio was a factor in the analysis in Experiment 1, and so why do the authors say this variable wasn’t considered?

Line 867-869: “the study considered only a subset of horizonal and vertical trajectories… rightward and upward movements were explored, while leftward and downward movements were not considered” The authors explicitly stated in Experiment 1 that the choice of a rightward direction was acceptable because “the choice of motion direction… does not exert any discernable impact on the resultant perception of causality (lines 385-387). If their earlier statement was true, then why is this limitation being admitted as important? Or to put it another way, if the authors knew this would a limitation, then why did they still choose to use just a limited number of directions?

Experimental design

Six Phases. I’m not entirely clear regarding the six phases. The first three and the sixth seem reasonably clear, but I’m confused about the fourth and the fifth. What happened in the fourth? Was the fifth based on a presumed recoding/reassignment of identity by the participant of the same display? Did participants have to extrapolate based on their memory for the previous stimulus, or was the stimulus shown again? It might be clearer if at the outset of the description there was a brief listing of the six phases with a parenthetical explanation of what took place in each phase. Could participants compete a phase six involving a recoding/reassignment of identity associations without having another phase two learning (that was tuned to the new coding/assignment)? What am I missing?

Agent/Patient Speed Ratio. I am confused about why this variable is included in the design and in the primary ANOVA (lines 381-382, 449-450) in Experiment 1. It involves agent speed, but agent speed is also a part of the agent speed variable, and thus the same stimulus parameter is being used in two different independent variables. I’m not entirely sure that wouldn’t violate an assumption of ANOVA. Why aren’t agent speed and patient speed treated as separate variables in the primary ANOVA? Then if the authors wanted, a subsequent one-way ANOVA on agent/patient ratio could be completed

Direction of Motion. Why was movement in Experiment 1 always toward the right? (lines 359-362, 366-368; 384-385). In Michotte’s day, and given his physical apparatus, a limitation to one direction was perhaps understandable, but in contemporary research in which the stimuli are computer animations, limitation to one direction does not make sense. It limits the generalizability of the responses unnecessarily (as we know there are differences in processing vertical motion relative to horizontal motion, oblique relative to cardinal axis, and leftward relative to rightward [even if the authors suggest those differences might not influence perception of causality (lines 385-387)] but they still list the number of directions as a limitation [lines 867-869]). Furthermore, the authors expressed an intent to reduce monotony (lines 378-380), and presenting different directions of motion across trials would certainty help in that goal.

Lines 246-248: This was confusing to me. After answering the first query regarding if they caused the motion of the stranger, did the participants watch the animation again while taking a different role before being asked whether the stranger caused their motion? Or did they answer both types of questions based on a single viewing (see General Issue B)? If the latter, how easy was it to assign a new perspective to the remembered image? I suspect (based upon literature on reinterpretation of visual images by Reisberg and colleagues) that it might not have been easy or even possible to have completely reinterpreted/reassigned the perspective of the original viewing.

Line 323: Based on Figure 1, readers will probably presume the square and circle were black, but the color should be explicitly stated (and on line 319, whether the background was light, medium, or dark gray should also be specified).

Lines 618-619: Isn’t animated motion characterized not just by changes in direction but also by changes in velocity (e.g., Tremoulet & Feldman, 2000)? Keeping velocity constant would thus seem to operate against a perception of animacy.

Lines 626-635: Viewing this figure seems more like looking down on a flat horizontal surface, whereas viewing the figures for a typical launching effect seem more like looking at a vertical picture plane. Were the participants instructed about the orientation of the stimulus? What about effects of orientation of the stimulus (e.g., effects of gravity)? How might such a difference potentially influence judgments of causality?

Validity of the findings

Perception vs. Inference. The authors might consider Firestone and Scholl’s (2016) paper in Behavioral and Brain Sciences that examines the possibility of top down effects in perception. Those authors provide a very specific list of requirements, and by comparing findings on whether phenomenal causality meet or do not meet the requirements for top-down effects, the authors of the current manuscript might be able to make a stronger case regarding whether physical or social causality involves only bottom-up perception or also involves top-down inference. Even so, isn’t the finding that the same chasing (social causality) display led to very different ratings in Experiments 1 and 2 evidence against a direct perception of causality and evidence in favor of an inference account (with inference being influenced by the other types of stimuli presented on other trials)? If social causality were truly a low-level perception, then shouldn’t the chasing (i.e., social causality) stimulus always have been perceived as a chasing (i.e., social causality) stimulus regardless of whatever stimuli might have been presented on other trials? If not, then why not?

Different Types of Causality. The authors suggest that participants were distinguishing between different types of causality in Experiments 1 and 2, but the instructions to participants don’t seem to have distinguished between physical and social causality. From the participant’s point of view, they were judging causality, and whether that was a physical or a social causality was (presumably) not explicitly considered by the participants. I’m not completely convinced the data warrant the claims regarding different types of impressions when the participants were not instructed about these different types (cf line 759-760). I suppose it might be argued that the different patterns across Experiments 1 and 2 (especially for the chasing display which the authors suggested considered as physical in Experiment 1 but social in Experiment 2) support the claim of different types of judgments, but if so, that raises the issue that judgments might reflect inferences based on other types of trials rather than a low-level perception per se

Line 80: Do A and B really have to be animate entities? I can imagine non-animate entities engaged in pursuit (e.g. a heat-seeking missile changing course to follow changes in the direction of motion of its target) and non-animate entities trying to avoid capture or avoid being hit. Must intentional equal animate? More generally, it seems like a non-animate artificial intelligence might be either a predator or a prey. With the technology in Michotte’s day this separation of animate and inanimate might have made sense, but does it still make sense?

Lines 252-254: Is this symmetrical? If the self is considered an active agent, then an internal or an intrinsic source of movement would be attributed to the self. Given this, then wouldn’t a patient to whom any causal power is attributed also have to possess movement? It seems like that would logically follow, but cases of a modified launching effect in which object B doesn’t move and object A is stopped or shattered lead to high causal ratings for the “patient”.

Lines 735-736: The lack of an effect of identity ratings seems somewhat inconsistent with the strong spontaneous assignments of identities and roles to stimuli in Heider--Simmel displays exhibited by most viewers. Perhaps this should be addressed.

Lines 754-756: If causality was directly perceived (i.e., based on low-level factors), then the same display in Experiments 1 and 2 should have received the same rating because the low-level features were the same in both instances. The fact that this display didn’t receive the same ratings in the two experiments suggests that inference based on other types of stimuli influence that judgment. If the authors feel this argument is incorrect, then they need to address this within the manuscript.

Additional comments

Line 75: For readers accustomed to APA style (which would probably be most of the readers of this article), seeing “&” in the main text is distracting. I would suggest the authors use “and” in the main text and limit “&” to parenthetical citations and the References section. Although only flagged here, this comment holds for the entire manuscript.

Lines 134-136: How is this known? In the absence of an explicit judgment, how is it known that the participants are responding to social causality (rather than to physical casualty or even to correlation)? This should be explicitly stated.

Line 164: What is the “these” referring to? Observers intuitive understandings? Or social causality judgments?

Line 230: Does “their” refer to the objects or to the individual’s actions? This should be rephrased.

Line 284-285: How? This should be described.

Line 311: Was “for” rather than “of” intended?

Line 326: “consequently” could be read as either subsequent in time or as a consequence of. These have different implications, and so this should be clarified.

Lines 784-785: Alternatively, might some force or motivation (or other non-kinematic) influence have been attributed to either of the stimulus objects?

Lines 907: I think “conclusive” should be “conclusions”

Line 921: Given the weakness of the effect, I’d say “is consistent with” rather than “suggests”.

References: Cited in the Review:

Bloom, P., & Veres, C. (1999). The perceived intentionality of groups. Cognition, 71(1), B1–B9
Gilden, D. L. (1991). On the origins of dynamical awareness. Psychological Review, 98(4), 554–568
Gilden, D. L., & Proffitt, D. R. (1989). Understanding collision dynamics. Journal of Experimental Psychology: Human Perception and Performance, 15(2), 372–383
Hubbard, T. L. (2013a). Phenomenal causality I: Varieties and variables. Axiomathes, 23, 1-42.
Hubbard, T. L. (2013b). Phenomenal causality II: Integration and implication. Axiomathes, 23, 485-524.
Hubbard, T. L., & Ruppel, S. E. (2013). Ratings of causality and force in launching and shattering. Visual Cognition, 21, 987-1009
Hubbard, T. L., & Ruppel, S. E. (2017). Perceived causality, force, and resistance in the absence of launching. Psychonomic Bulletin & Review, 24, 591-596.
Proffitt, D. R., & Gilden, D. L. (1989). Understanding natural dynamics. Journal of Experimental Psychology: Human Perception and Performance, 15(2), 384–393.
Tremoulet, P. D., & Feldman, J. (2000). Perception of animacy from the motion of a single object. Perception, 29(8), 943–951
White, P. A. (2006). The causal asymmetry. Psychological Review, 113(1), 132–147.
White, P. A. (2007). Impressions of force in visual perception of collision events: A test of the causal asymmetry hypothesis. Psychonomic Bulletin & Review, 14(4), 647–652.
White, P. A. (2012). The experience of force: The role of haptic experience of forces in visual perception of object motion and interactions, mental simulation, and motion-related judgment. Psychological Bulletin, 138, 589–615

---

## Round 0.2 · Minor Revisions

I sent the revised version of your manuscript to the same reviewers of the first round: as you will see, Reviewer 1 is now satisfied (only has a couple of very minor comments), while Reviewer 2 still has some concerns - not as strong as in the first round but to be carefully addressed anyway.

·

Basic reporting

The revision is, like the previous draft, well-written and clear. One small correction: line 922, the "and" should be retained, as well as the comma. A list of two that includes a compound item is a bit of a stylistic headache, but I believe that would be the clearest.

Experimental design

The design remains excellent.

Validity of the findings

My concerns have been addressed. The current version of the manuscript appropriately constrains its conclusions and offers insightful commentary on the issues raised in the previous review.

One small comment is that I don't think I understand the second manipulation described on lines 977-980. To me it sounds like the same thing as the first manipulation, and I'm not sure how it would imply self-initiated motion if it is controlled by the keys. A little clarification on what the intended manipulation would look like would be helpful.

Additional comments

No further comments.

Reviewer 2 ·

Basic reporting

Line 1: The title could be potentially misleading, as participants were not told to judge physical causality or to judge social causality, and in fact, the instructions regarding causal judgments were the same for both experiments. Whether participants judged physical causality or social causality was inferred, and while such inferences seem plausible, there is no guarantee that they are necessarily correct.

Lines 55-56: How are things such as “trajectories, speeds, distances and timing” examples of subtle visual features? I would have thought they were usually obvious visual features. What is meant by “subtle” in this context?

Line 65: Unless the authors want to insert “effects” after “behavioral” on line 63 (which would parallel their writing on lines 145-148), they should delete the “effects” on line 65 and insert “effects” before “that can” on line 66. Regardless of which version is chosen, the style should be consistent.

Line 82: The phrase “imparting the scene with the percept…” implies that it is the scene, rather than the observer, that has the percept.

Line 102: I suspect “speed” should be “speeds” as A and B could have different speeds.

Line 147: Paroval and Guidi (2020) is cited in the main text but is not included in the References section.

Line 165: Gemelli and Cappellini is listed as published in 1958 in the main text but as published in 1959 in the References section.

Line 201: I don’t understand what “stemming with kinematic cues” means. Perhaps the authors meant “stemming from kinematic cues”?

Line 223: I would consider inserting a comma after “interactions”.

Lines 227-228: I would suggest the authors use more parallel construction, as that is generally considered a better style for scientific and technical writing, and usually helps the reader understand more easily. In the example I’ve flagged here, more rapid movement “is construed” but slower movement “assumes the role”. Either “is construed” or “assumes the role” should be used in both instances. I’ll only flag this one instance, but I suggest the authors check their manuscript for other instances in which parallel construction might be used (e.g., lines 448-449 introducing the fourth and fifth phases could be made more parallel to lines that introduced the first, second, and third phases; practice trials were referred to “the practice block” in Experiment 1 [line 394] but as “a trial block” in Experiment 2 [line 712]; etc.).

Line 233-241: The authors just discussed differences between physical causality and social causality, and then they move into a discussion of causal asymmetry. However, hasn’t discussion of causal asymmetry generally been limited to physical causality? The way this discussion s structured, readers will conclude that White’s discussion of causal asymmetry involved both physical causality and social causality, but if memory serves, it doesn’t,

Lines 374-399, 403-436: These paragraphs are a bit long. It might be helpful for some readers if they were divided into smaller paragraphs. As long as the transition to a new phase is clear, it shouldn’t be necessary to cram all of the information regarding a specific phase into a single paragraph.

Lines 988-1017: Given how much of the text of the Conclusions section is strike-through, it might be better to collapse the remaining text into a single paragraph rather than three really short paragraphs.

Line 1046: There’s an extra comma on this line.

Line 1219: I suspect “Shultz” should be “Schultz”.

Experimental design

Lines 362-365: Were the matching stimuli in the fourth phase and the “reencountered” animations in the fifth phase presented in the same order as that in which participants had viewed those stimuli in the second and third phases, respectively? It might be useful if mention of what manipulation was being checked was included within the sentence describing the sixth phase.

Lines 421-422: “Both the agent and the patient remained stationary for 500 ms before disappearing from the screen” suggests to me that the agent vanished before the patient (after all, the patient only starting moving within -167 to +233 of the time of when the agent stopped moving). Thus, if both agent and patient vanished 500 ms after each became stationary, the agent would vanish before the patient vanished. What I suspect the authors meant was that both agent and patient vanished 500 ms after the patient stopped moving. Either way, this needs to be clarified.

Lines 435-436: Prior studies might not have found an effect of direction on judgments, but those studies did not look at the effect of the variables of interest in the current study (e.g., self-identification, etc.). Just because a variable such as direction might not have a general effect in a previous study of any given topic does not mean that variable might not have an effect on a previously unstudied manipulation regarding that topic. I know such a comment won’t have any effect on the current manuscript, but I still suggest it be kept in mind for future studies.

Lines 690-691: Was there any assessment of whether the two black rectangles had the desired effect of making the animation look like it took place on a stage (e.g., did a different group of participants describe their perception of the stimulus or choose which of several potential descriptions best matched their perception)? If the display is perceived as vertical (upright), which is how I think the authors wanted it to be perceived, that might lead to a different account for the findings than if the animation were perceived as looking down on a flat horizontal surface (e.g., moving “up” rather than moving “to the left”). Perception of causality is related to naïve physics, and an analogous case in naive physics literature involves a C-shaped or spiral tube (i.e., when a drawing of such a stimulus was displayed, interpretation of the responses could be influenced by whether then stimulus was thought to be standing upright or lying flat on a surface). I’m not suggesting such an effect of orientation necessarily took place, but I do think such a possibility cannot just be dismissed, and it is possible that participants did not necessarily interpret the stimulus as the authors thought they should.

Lines 788-790: It would be helpful for readers if the authors could state why different priors were used.

I’m not entirely sure I agree with the authors’ response to my concerns about including both agent (absolute) speed as well as agent/patient speed ratio in the same ANOVA. I certainly do not dispute their claim that agent/patient speed ratio influences perception of causality, but that doesn’t really address my concern that agent speed is included in two different variables (and so those two variables aren’t independent and would be expected to be correlated). I’m not convinced by their comments in the cover letter that “agent speed” and “speed ratio” could alternatively be labeled as “absolute speed’ and “relative speed”; yes, the speed ratio could be labeled as relative speed, but to call the agent speed “absolute speed” seems to ignore that the patient also has an absolute speed that can influence perception of causality. In theory, at least, the agent and patient speeds do not have to be correlated in any set of stimuli, as agent speed and patient speed could potentially be fully crossed in the design. Perhaps an ANCOVA might be appropriate?

Validity of the findings

I’m not completely convinced the experiment clearly distinguishes between judgments of physical causality and judgments of social causality. In both Experiments 1 and 2, the participants were asked “How much did YOU cause the motion of the STRANGER” or “how much did the STRANGER cause YOUR motion?” Use of “YOU” and “STRANGER” might be supposed to evoke social causality, and if so, it isn't clear why physical causality would be involved. These instructions do not mention whether physical causality should be judged or social causality should be judged. The authors demonstrate that certain aspects of agent and patient motion do indeed influence judgments of a generic “causality”, but whether those influences correspond to differences between physical casualty and social causality is not necessarily demonstrated. At the least, this should be more nuanced.

The statements that “The results of Experiment 1 suggest that animations with distinct visual cues of physical causality tend to bias explicit judgments toward physical causality…” (lines 600-601) and “may have shifted participants response criterion toward physical causality” (line 897) seem appropriately cautious. However, in other places the authors use much stronger language (e.g., “exert minimal or no impact on explicit judgments of social causality” [lines 873-874], “would lead to higher ratings of physical or social causality” [line 884], “influence judgment of social causality” [line 908], “judgments of social causality [line 912], “the distinct interpretation of ‘causality’ between the two experiments” [line 955]) regarding the type of judgments, and they appear to assume the type of causality being judged.

Additional comments

Line 216, 994: The term “paradigm” is incorrectly used. A paradigm refers to the axioms, acceptable questions, and acceptable methods of investigation for studying a large area or domain (such as a discipline). Examples of paradigms within psychology include behaviorism, information processing, and connectionism (neural networks). A “paradigm” is not the specific design or methodology of a given study or paper. I admit this is an incorrect usage that is common in the literature (and in some of the papers the authors cite), but it is still incorrect, and prior incorrect usage doesn’t justify continued incorrect usage.

Lines 253-255: It might be worth mentioning that this is a common explanation for how superstitions arise.

---

## Round 0.3 · accepted · Accept

I commend the authors for having addressed the additional comments received in the second round of review. I have assessed the revision myself, and I'm happy with the current version; in my opinion, the manuscript is ready for publication as it is.